# Incorporating human dimensions is associated with better wildlife translocation outcomes

Mitchell W. Serota [1] ✉, Kristin J. Barker [1], Laura C. Gigliotti[1], Samantha M. L. Maher[1], Avery L. Shawler [1], Gabriel R. Zuckerman [1], Wenjing Xu[1], Guadalupe Verta [1], Elizabeth Templin [1], Chelsea L. Andreozzi [1] & Arthur D. Middleton[1]

Wildlife translocations are increasingly used to combat declining biodiversity worldwide. Successful translocation often hinges on coexistence between humans and wildlife, yet not all translocation efforts explicitly include human dimensions (e.g., economic incentives, education programs, and conflict reduction assistance). To evaluate the prevalence and associated outcomes of including human dimensions as objectives when planning translocations, we analyze 305 case studies from the IUCN's Global Re-Introduction Perspectives Series. We find that fewer than half of all projects included human dimension objectives (42%), but that projects including human dimension objectives were associated with improved wildlife population outcomes (i.e., higher probability of survival, reproduction, or population growth). Translocation efforts were more likely to include human dimension objectives if they involved mammals, species with a history of local human conflict, and local stakeholders. Our findings underscore the importance of incorporating objectives related to human dimensions in translocation planning efforts to improve conservation success.

Over the last century, more than 200 vertebrate species have gone extinct, and many more have experienced range contractions, extirpations, and population declines[1]. Wildlife translocation, defined here as the intentional movement of organisms from one site to another for the benefit of conservation serves as an increasingly important tool to combat widespread declines in global biodiversity[2–5]. However, wildlife translocation programs have been met with mixed results. High-profile wildlife translocation success stories include the reintroduction of Arabian oryx (*Oryx leucoryx*) throughout the Arabian Peninsula and the peregrine falcon (*Falco peregrinus*) throughout the United States[6–8]. Conversely, reintroduced populations of brush-tailed bettongs (*Bettongia penicilliata*) in Australia and red wolves (*Canis rufus*) in the United States swiftly declined to unsustainable levels[9,10]. Translocation programs require considerable time and resources, and their failure

can lead to distrust between stakeholders, the loss of resources, and even the extinction or extirpation of entire populations or species[4,11]. Thus, understanding why some efforts succeed where others fail is key to designing future wildlife translocation programs and allocating scarce conservation resources. To date, such understanding has remained elusive, likely due in part to the underreporting of conservation struggles relative to successes[12,13].

Investigations into common drivers of wildlife translocation success have largely focused on biological and ecological factors such as climate suitability, reintroduction site quality, source population origin, and the number of reintroduced individuals[5,14–16]. However, as conservation efforts increasingly occur in landscapes shared by humans and wildlife, the success of translocations has become more reliant on coexistence with people[17]. Therefore, human dimensions, or

[1]Department of Environmental Science, Policy, and Management, University of California - Berkeley, Berkeley, 130 Mulford Hall, Berkeley, CA 94720, USA. ✉e-mail: mitchell_serota@berkeley.edu

the social, political, psychological, economic, and cultural components of conservation, are increasingly recognized as critical to the success of wildlife translocations[18–21]. Human dimension-related activities in wildlife conservation can be either foundational (providing information needed to understand the local context and stakeholders) or functional (being directly applied to management issues)[22].

Incorporating human dimensions may ultimately prove as important to achieving conservation goals – if not more important - than biological or environmental factors, because most threats to wildlife are directly attributed to humans[23]. Indeed, human dimensions have informed the design of translocations across multiple taxa including fish[24], mammals[25,26], birds[27,28], reptiles[29], and amphibians[30,31]. Examples include resource provisioning to protect livestock from translocated wildlife, education programs in local communities and schools, media campaigns to influence attitudes towards wildlife, economic benefits for landowners living with wildlife, and legal enforcement against illegal wildlife trade. Many groups working to reintroduce wildlife now integrate social and ecological information into their conservation plans to better predict areas of wildlife tolerance, potential conflicts, and the distribution of benefits to local communities[32–35]. In the IUCN Guidelines for Reintroductions and Other Translocations, the inclusion of human dimensions is considered integral to the design, implementation, and evaluation of translocations[36]. However, despite the recognized importance of human dimensions, these factors are still largely missing from many conservation initiatives[21,37,38]. Potential explanations for this gap includes scarcity of resources, administrative and funding legacies, and lack of interdisciplinary collaborations[37]. Although many individual case studies highlight the importance of including human dimensions in the design and implementation of wildlife translocation programs, overarching relationships between translocation success and human dimension considerations have not been comprehensively evaluated.

To identify relationships between the inclusion of human dimension objectives in wildlife translocation efforts and program outcomes, we synthesized information from case studies reported in the International Union for Conservation of Nature (IUCN) Global Re-Introduction Perspective Series[39–45]. First, we tested whether setting human dimension objectives increased the probability of a positive outcome (i.e., widespread survival, reproduction, or population growth) for the translocated wildlife population. Second, we identified the primary factors influencing whether translocation efforts set human dimension objectives. We predicted that the probability of including human dimensions in project objectives would be higher (a) for projects translocating wider-ranging taxa like mammals and birds whose broad ranges often overlap with human-influenced areas, (b) in areas where the key threats to the species were locally attributed to humans, (c) where humans have experienced conflict with the species of interest, and (d) when local stakeholders played an active role in the project (Table S1). Additionally, given increasing attention to human dimensions in conservation and their explicit recommendation in the IUCN Guidelines for Reintroductions and Other Translocations published in 2013, we predicted that the inclusion of human dimension objectives would increase over time.

We found evidence that explicitly setting objectives related to human dimensions was associated with an increased probability of a positive outcome for the translocated wildlife population. However, fewer than half of all case studies included human dimension objectives when planning their translocation. Translocation efforts conducted without including local community members, for example those led solely by academic institutions, governments, non-profits, or zoos, were less likely to have a positive outcome. The probability of setting human dimension objectives was higher for restoration efforts of mammals and birds and for species with local threats directly related to humans or a reported history of human-wildlife conflict.

Promisingly, the inclusion of human dimension objectives in wildlife translocations has increased over time. Our results underscore the importance of human dimensions in wildlife translocation success, revealing that translocations and conservation efforts benefit from incorporating human-related factors along with biological and environmental considerations.

## Results

We analyzed 305 case studies of wildlife translocations from 7 IUCN reports published between 2008 and 2021. Translocations occurred from 1922 to 2018 and included 121 mammals, 77 birds, 40 fish, 40 reptiles, and 27 amphibians. Most case studies occurred in North America ($n = 69$) and Asia ($n = 67$), followed by Oceania ($n = 56$), Europe ($n = 53$), Africa ($n = 35$), and South America ($n = 25$). Of the 305 case studies, 127 case studies (42%) included human dimension objectives in either their Goals or Success Indicators. One hundred and seventy-three case studies (57%) included human dimensions in either their Lessons Learned or Major Difficulties Faced, 76 of which (43%) did not include human dimension objectives in their Goals or Success Indicators. Most projects resulted in a positive outcome ($n = 272$); approximately 11% ($n = 33$) reported a negative outcome. Overall, translocation efforts that included human dimension objectives were significantly more likely to have a positive outcome (0.94; 95% CI = 0.88–0.97) than the translocation efforts that did not include human dimension objectives (1.02, 95% CI = 0.07–2.10; $p < 0.01$). Both project time length and taxa were insignificant ($p > 0.05$ in both cases).

Of the six key strategies we identified for including human dimension objectives, education was the most common ($n = 111$), followed by engaging locals ($n = 96$), providing economic benefits ($n = 41$), increasing social tolerance ($n = 32$), enforcing regulations ($n = 19$), and supplying cultural benefits ($n = 9$). The inclusion of human dimension objectives varied significantly between taxonomic groups, threats to the species, the groups involved in the restoration, and whether the authors reported a history of human conflict. Across taxonomic groups, translocation efforts of both mammals (0.53; 95% CI = 0.44–0.62) and birds (0.41; 95% CI = 0.31–0.53) had a significantly higher probability of including human dimension objectives than amphibians (0.15; 95% CI = 0.06–0.34) ($p < 0.01$ and $p = 0.01$, respectively; Fig. 1). Translocation efforts of mammals also had a higher predicted probability of including human dimension objectives compared to fish (0.33; 95% CI = 0.20–0.48; $p = 0.02$; Table S3). Case studies that reported a history of conflict with the species had a

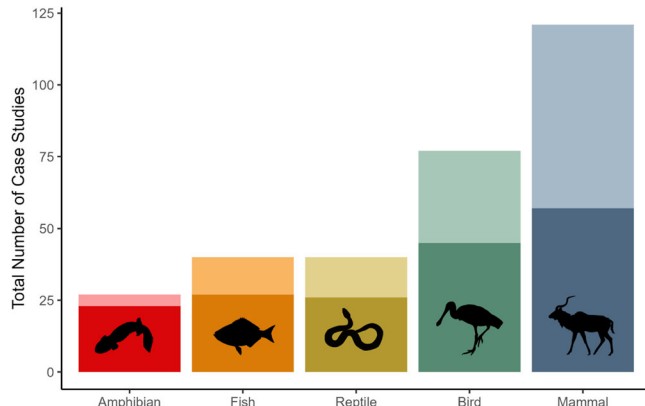

**Fig. 1 | Inclusion of human dimension objectives in wildlife translocations varied among taxa.** Data is based on case studies from the IUCN Global Reintroduction Perspectives Series (2008–2021). Lighter shading indicates the number of case studies that included human dimension objectives; darker shading represents case studies that did not include human dimension objectives. By taxon, the percent of translocations that did not include human dimension objectives were: Amphibians: 85%; Fish: 68%; Reptiles: 65%; Birds: 58%; Mammals: 47%.

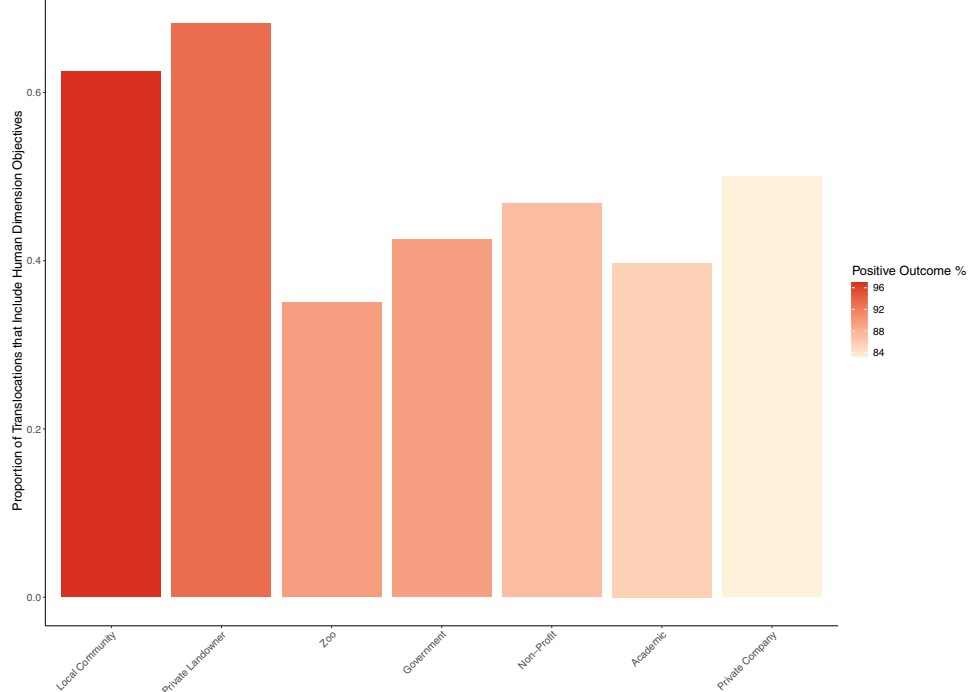

**Fig. 2 | Active inclusion of local stakeholders is linked to improved translocation outcomes.** Bars indicate the proportion of studies reported in the IUCN Global Re-introduction Perspectives Series (2008–2021) that incorporated human dimension objectives in their restoration project varied based on the types of groups involved in the project. The color gradient from lighter red (lower) to darker red (higher) represents the percentage of studies involving each group that had positive translocation outcomes, regardless of whether human dimension objectives were included. By group type, the percent of translocations that resulted in a positive outcome were local community: 97%; private landowner: 93%; zoo: 89%, government: 89%; non-profit: 87%; academic: 85%; private company: 83%.

predicted probability of including human dimension objectives of 0.62 (95% CI = 0.50–0.73), significantly higher than the predicted probability of including human dimension objectives for translocation efforts of a species without a history of conflict of 0.36 (95% CI = 0.30–0.42; $p < 0.01$).

Translocation efforts that involved local communities (0.63; 95% CI = 0.50–0.73) and private landowners (0.68; 95% CI = 0.53–0.80) were significantly more likely to include human dimension objectives than restoration efforts that involved academics (0.39; 95% CI = 0.32–0.48), zoos (0.35; 95% CI = 0.26–0.46), government agencies (0.42; 95% CI = 0.36–0.49), nonprofits (0.47; 95% CI = 0.39–0.54), and private companies (0.50; 95% CI = 0.31–0.69) ($p < 0.05$ in all cases, Fig. 2, Table S4). Translocation efforts that involved local communities had a significantly higher predicted probability of a positive outcome (0.97, 95% CI = 0.88–0.99) than translocation efforts that involved academics, non-profits (0.87; 95% CI = 0.81–0,91), and private companies (0.83; 95% CI = 0.63–0,93). Finally, case studies where the species was threatened by transportation and service corridors, energy production or mining, agriculture or aquaculture, and biological resource use had the highest predicted probability of including human dimension objectives, whereas translocation efforts where the species was threatened by climate change, invasive species, and natural system modifications had the lowest predicted probability of including human dimension objectives (Table S4).

After we identified taxonomic groups, stakeholder groups involved in the translocation, IUCN threats, and a local history of conflict as significant predictors of the inclusion of human dimension objectives, we evaluated the relative importance of each predictor in a global model. Like the univariate model results, whether the species was a mammal, local history of conflict, and whether the translocation involved local community groups were all significant predictors of including human dimension related objectives (Table S6). However, the translocation of fish taxa and the presence of a direct human threat were no longer significant when considered in conjunction with the other variables ($p > 0.05$ in both cases; Table S6).

Translocation efforts from the IUCN case studies spanned from 1922 to 2018. However, because the case study from 1922 was an outlier occurring 38 years before any other case study in the dataset, we removed it from the temporal analysis and began instead with a case study from 1960. Since then, the inclusion of human dimension objectives has increased over time from an estimated probability of inclusion of 0.20 (95% CI = 0.09–0.40) in 1960 to an estimated probability of inclusion of 0.50 (95% CI = 0.40–0.60) in 2018 ($p = 0.05$; Fig. 3). However, there was no significant increase in the inclusion of human dimension objectives following the publication of IUCN Guidelines for Reintroductions and Other Conservation Translocations (before publication, $n = 248$; after publication, $n = 38$; $p > 0.05$).

## Discussion

Human dimensions are increasingly thought to play a critical role in the success of conservation efforts, and our work supports this assertion by quantifying a strong relationship between the inclusion of human dimension objectives and the probability of success for wildlife translocation projects. Our analysis of all vertebrate case studies reported in the IUCN Global Re-introduction Perspectives Series from 2008 to 2021 revealed projects that included human dimension objectives during the planning process were associated with a 10% higher probability of a positive outcome (i.e., survival, reproduction, and/or growth of a wildlife population) for the wildlife population than those that did not. Our findings therefore reveal opportunities to improve the outcomes of wildlife translocations not only by addressing the environmental and programmatic factors known to influence conservation success, but also by addressing human dimensions through facilitating education opportunities, providing economic benefits, engaging locals in conservation, increasing social tolerance, improving cultural benefits, or enforcing regulations. Our analysis also

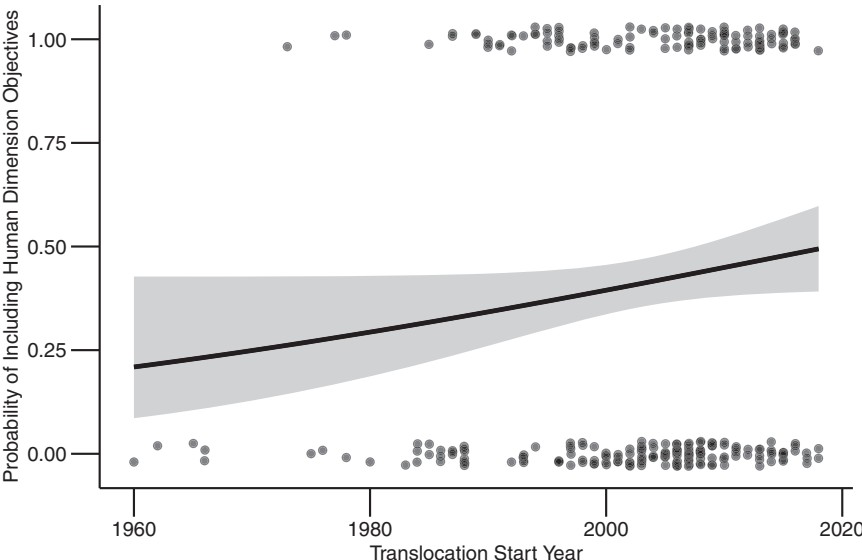

**Fig. 3 | The inclusion of human dimension objectives has increased over time.**
The line indicates the predicted probability of including a human dimension
objective in a wildlife translocation through time; the shaded area represents the
95% confidence interval, and the points indicate raw data (binary inclusion or
exclusion of human dimension objectives). We applied jitter to the points to
increase readability. Data source: Global Re-introduction Perspectives Series
(2008–2021).

highlights the importance of engagement and collaboration with local
stakeholders by traditional wildlife conservation groups.

Our results suggest that the inclusion of human dimension
objectives is biased towards translocations of mammals and, to a lesser
extent, birds. It has long been suggested that there is a taxonomic bias
towards mammals and birds in conservation research, despite
amphibians being more threatened and declining more rapidly than
both birds and mammals[46,47]. A recent analysis identified agriculture,
logging, and hunting as the most common threats for amphibians
globally, all of which are directly caused by humans[48]. Even so, few
amphibian restoration efforts planned for human dimensions, perhaps
due to a lower perceived value of this taxa to natural ecosystems and
society[49]. There may be a number of reasons why translocations of
mammals and birds are more likely to incorporate human dimensions.
In general, mammals and birds are larger and wider ranging than other
taxa, putting them at a greater risk of conflict with humans. Con-
servationists might be more attuned to this risk, and therefore more
likely to include human dimension objectives in related translocation
efforts[50]. In addition, methods for including human dimensions like
education programs and directly involving community members in
restoration efforts might be more straightforward for species con-
sidered "charismatic," which tend to be larger mammals.

Conservationists have long called for more collaborative and
bottom-up approaches, like community-based conservation, which
center conservation around the needs and wants of local
communities[51]. In addition, there is a growing recognition of the value
of acknowledging, learning, and integrating critical ecological knowl-
edge of local communities and indigenous groups[52]. In some cases,
top-down approaches in wildlife conservation have led to the dis-
placement of local people and increased economic inequality, while
providing little to no benefit for local people or even wildlife or eco-
systems more broadly[53,54]. These negative experiences may sow dis-
trust and build local resentment to conservation efforts, thereby
damaging long-term conservation success[55]. Conversely, bottom-up
approaches that democratize conservation and prioritize the needs
and knowledge of local communities can lead to increased trust,
learning, and better outcomes for wildlife conservation[55–57]. Still, many
of the translocation projects we reviewed did not include local com-
munity groups.

While our results provide clear support for the consideration of
people in wildlife translocations, not all human-focused conservation
strategies led to better outcomes for wildlife populations. Although
conservation-related regulations can serve as an effective tool for
improving translocation outcomes, some instances of militarized
enforcement has created repressive and violent policies that under-
mine biodiversity conservation by further alienating local
communities[58]. Additionally, while ecotourism and other economic
incentives can yield positive conservation results, they can also cause
tension among community members around issues of inequitable
benefit sharing, ultimately undermining conservation objectives[59].
Therefore, the implementation of human dimension objectives must
carefully consider all possible social and ecological outcomes, and
interdisciplinary science may be key to future restorations.

Only 42% of case studies reported in the IUCN Global Re-
introduction Perspectives Series reported human dimension objec-
tives in the planning phase of their projects. Over the last few decades,
there have been significant calls to better link conservation goals to
sustainability goals, as well as to human values and wellbeing[60–64].
Additionally, major national and international conservation initiatives
like the Convention on Biological Diversity and California's 30 × 30
Executive Order aim to center human dimensions in their respective
frameworks[65,66]. The IUCN Guidelines for Reintroductions and Other
Conservation Translocations also has important recommendations for
evaluating the social feasibility and conducting socioeconomic risk
assessments of translocations[67]. These advancements have all likely led
to the observed increase in reported human dimension objectives.
Still, even in the most recent 2021 IUCN report, only 50% of reported
translocations set human dimension-related Goals or Success
Indicators.

Further highlighting the importance of human dimensions in
wildlife translocations, 57% of case studies cited human dimensions as
a Lesson Learned or Major Difficulty. In fact, an additional 15% of case
studies included human dimensions in either their Lessons Learned, or
Major Difficulties Faced compared to their Goals or Success Indicators.
In Ireland, inadequate consideration and consultation of a rural farm-
ing community prior to the reintroduction of the white-tailed eagle
(*Haliaeetus albicilla*) resulted in widespread poisoning and high eagle
mortality[68]. Future translocations should utilize conservation planning

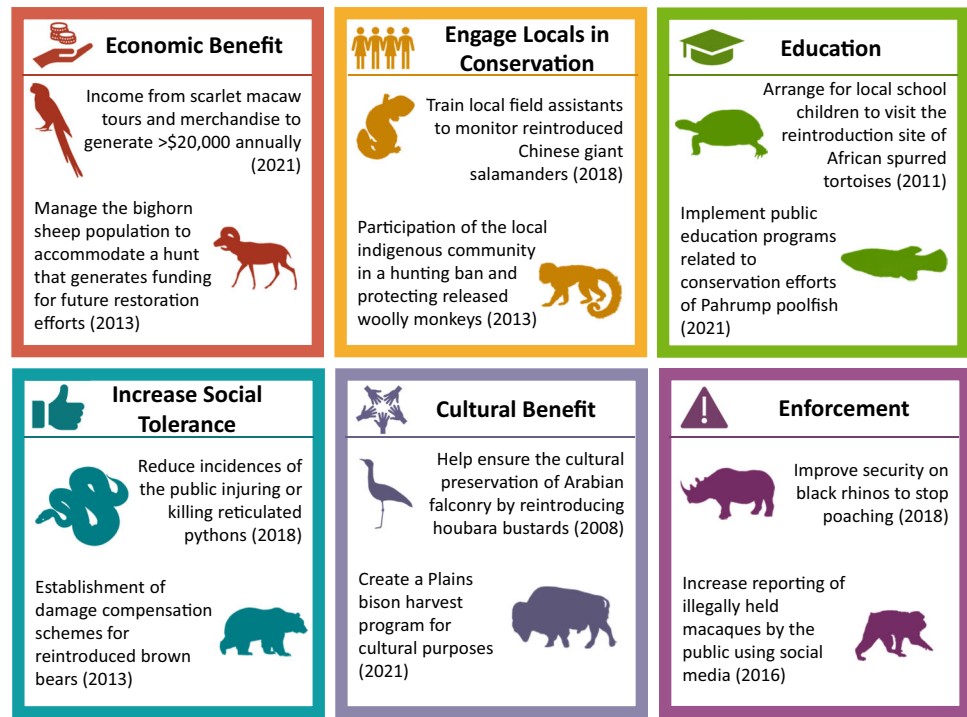

**Fig. 4 | Strategies used to incorporate human dimensions in wildlife translocation.** Strategies were identified based on human dimensions reported in project goals or success indicators from case studies in the IUCN Global Re-Introduction Perspective Series; the figure includes key examples from each strategy[39,41–45].

tools that integrate both ecological and socioecological variables which have been found to better predict the expansion of recolonizing wildlife populations, the occurrence of human-wildlife conflict, and effective release sites for reintroduced individuals[32,33,69].

Education and outreach were the most commonly reported human dimension strategy incorporated in translocation project planning. Importantly, education and outreach can help introduce people to the species and the goals of the project, as well as influence the behavior of the general public[70]. For example, conservationists who reintroduced the critically endangered Pahrump poolfish (*Empetrichthys latos*) in Nevada largely attributed their success to increased public buy-in following an education and media campaign[45] (Fig. 4). Other popular human dimension objectives include increasing social tolerance and providing economic benefits to aid biodiversity conservation efforts[17,71]. In Chile, wildlife tourism of an increasing puma (*Puma concolor*) population has led to a sharp decline in support for the lethal control of pumas, the primary cause of their decline and extirpation throughout the region[72]. Interestingly, enforcement was one of the least commonly reported strategies despite increasing global attention to anti-poaching and wildlife trafficking efforts[58]. Future analyses that further disentangle the effectiveness of various strategies may aid in increasing the implementation of human dimensions by conservation organizations.

Our analysis is just the beginning of better understanding how human dimensions impact the success of wildlife restorations globally. We note that our binary classification of outcomes solely focuses on the outcome to the population of the species as stated by the authors, so it does not account for success related to knowledge gained for future restorations, stakeholder support, or other non-population-related successes. Further, our results may be influenced by reporting bias against translocations conducted by smaller organizations as well as translocation failures. The publication rate for successful translocations is likely to be higher as many failed translocations are underreported[73], which may partially account for the low failure rate (11%) in the IUCN report. Thus, our analysis is representative of the literature, but not all attempted translocations. Still, we've found that

major, well-resourced conservation organizations and relatively over-reported successful translocations are failing to incorporate human dimensions into their efforts; this speaks particularly strongly to the overall lack of consideration for human dimensions if arguably the best-resourced and most successful translocations are foregoing important opportunities to improve conservation outcomes and local partnerships.

Effective wildlife translocation clearly requires thoughtful consideration of the human dimensions that make conservation projects more sustainable and successful. As biodiversity continues to decline, there is an urgent need to integrate well-established biological and environmental schema with a deeper understanding of the social and human dimensions that will help to avoid unaffordable failures. A single wildlife restoration failure can result in a species' extinction[4,11], as well as the loss of millions of dollars and the sowing of distrust between communities and conservation institutions. Therefore, analyses to understand even marginal gains in translocation success can be impactful for future conservation efforts. Tools and practices to better understand the human dimensions of translocations like stakeholder engagement or participatory approaches can be both costly and time consuming, yet our study underscores their importance. While the literature is ripe with frameworks and guidelines that emphasize the need to include human dimensions, too few projects adequately plan for the human dimensions needed for long-term success[36,74].

## Methods
### IUCN Global Re-introduction perspectives series
The IUCN Global Re-introduction Perspectives Series publishes conservation translocation case studies of plants, invertebrates, amphibians, reptiles, birds, fish, and mammals from around the world[39–45]. The goal of the series is to provide a global synthesis of the challenges facing biodiversity translocation projects. The series has published 7 volumes from 2008 to 2021. All case studies share the same structure with the following sections: Introduction, Goals, Success Indicators, Project Summary, Major Difficulties Faced, Major Lessons Learned, and a self-evaluated ranking of the success of the project with a section

on the Reason(s) for Success. Participants in the series are given a blank template and a few examples of case studies from previous volumes to draw upon. The format provides a standardization not otherwise possible with traditional literature reviews. However, the abbreviated format and self-reporting nature likely does not encompass every detail of the translocation, nor does the collection of case studies chronicle every wildlife translocation attempt. For this analysis, we focused on case studies of all vertebrates ($n = 305$). In total, there were 268 unique species in the dataset.

## Data collection and categorization

For each case study, we evaluated whether each section in the report (Goals, Success Indicators, Major Difficulties Faced, and Major Lessons Learned) contained information related to the human dimensions of the translocation. We defined setting human dimension objectives as a binary yes/no based on whether the project explicitly included either a Goal or Success Indicator that related to any aspect of human-related cultural, political, economic, social, or psychological considerations[75]. We then identified six key strategies into which we categorized each human dimension related Goal or Success Indicator: providing education, engaging locals, increasing social tolerance, supplying economic benefits, enforcing regulations, and improving cultural benefits (Fig. 4). In addition, we recorded the location, start year of the project, groups or stakeholders involved in the translocation, threats to the species, and whether there was a history of conflict reported between that species and humans in the translocation area. The group(s) or stakeholder(s) for each translocation were identified from the authors' affiliations and the Project Summary of each case study and were classified as government, academic, zoo, non-profit, local community, private landowner, and private company. Stakeholder classifications were based on project involvement; therefore, many case studies included multiple groups. The threats to each species were classified according to the IUCN Red List of Threatened Species Database and included direct human threats (e.g., residential & commercial development, agriculture & aquaculture, energy production & mining, transportation & service corridors, biological resource use, human intrusion & disturbance, natural systems modification), and indirect human threats (e.g., invasive & other problematic species, genes & diseases, pollution, and climate change & severe weather)[76].

The success of wildlife translocations can be measured in multiple ways including changes to the target population, impacts to the ecosystem, and knowledge gained from the project. In the IUCN Global Reintroduction Perspectives Series, all authors rate the success of the project from 'Highly Successful' to 'Failure.' However, there may be inconsistency in how the authors of different projects define success. Therefore, we classified the outcome of the project as positive or negative based on the outcome to the wildlife population reported in the Project Summary and Reason(s) for Success sections of the reports. Case studies that we classified as having a positive outcome reported on a scale of widespread survival, reproduction, and/or population growth, whereas case studies classified as a negative outcome reported either widespread mortality or population extinction (Table S2). Therefore, case studies only needed to report a minimum of widespread survival of the translocated individuals to be classified as a success. We used a binary positive or negative outcome instead of each individual outcome to reduce bias from the species in the case study (e.g., differences in generation times) or project (e.g., length of project) which could greatly impact differences in the reported outcome (e.g., survival vs. reproduction). Further, the binary outcome also increased the repeatability in our assessment of the project due to the clear differences between positive (widespread survival, reproduction, or population growth) and negative (widespread mortality and population extinction) outcomes.

We classified human dimension objectives and wildlife population outcomes through a collaborative calibration process. First, each coauthor independently evaluated thirty case studies to identify broad classifications of human dimension strategies. Next, we worked together to synthesize and refine classifications to comprehensively cover all human dimensions reported. We then reviewed all projects using the classification framework ensuring consistency by discussing all potentially ambiguous classifications with the entire group.

## Analysis of human dimension objectives across wildlife translocations

We used a series of logistic regression models to test our predictions related to human dimensions. First, we assessed whether the inclusion of human dimension objectives affected the outcomes of wildlife translocations using a multivariate logistic regression model with the translocation outcome (positive or negative) as the response variable and whether the project included human dimension objectives as a binary predictor variable. We also included two factors that may impact the reported outcomes as a predictor variable: 1) project time length, because longer project time lengths could increase the probability of observing a positive outcome, and 2) taxa, since population increases could be more difficult to identify in taxa with longer generational times.

Next, we examined how the inclusion of human dimension objectives (as a binary response) varied among taxonomic groups, threats to the species, existence of a local history of conflict with the species, and the stakeholder groups involved in the translocation. We evaluated differences for each variable using Tukey's post-hoc pairwise comparisons. To compare the relative importance of each variable, we then conducted a multivariate logistic regression with the inclusion of human dimension objectives as the binary response and the taxonomic group, existence of a local history of conflict with the species, whether the translocation involved local community groups, and whether one of the listed IUCN threats was a direct human threat as covariates.

Finally, to evaluate changes in the inclusion of human dimension objectives over time, we used two separate univariate logistic regressions with the inclusion of human dimension objectives as a binary response variable. One model included the restoration start year as the predictor variable; the other included a binary variable representing whether the case studies occurred before or after 2014 to capture whether the inclusion of objectives related to human dimensions increased following the publication of the IUCN Guidelines for Reintroductions and Other Conservation Translocations in 2013. All statistical analyses were conducted in R version 4.0.2, and we defined statistical significance based on an alpha level of 0.05[77]. We generated figures using the 'ggplot2' package in R[78].

### Reporting summary

Further information on research design is available in the Nature Portfolio Reporting Summary linked to this article.

## Data availability

The dataset made from the IUCN Global Re-Introduction Perspective Series have been deposited with Zenodo under accession code https://doi.org/10.5281/zenodo.7086487. The IUCN Red List of Threatened Species and the IUCN Global Re-Introduction Perspective Series can be found at iucnredlist.org and iucn.org.

## Code availability

The code used for the data analysis used in this study has been permanently archived with Zenodo (https://doi.org/10.5281/zenodo.7438853).

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

## Acknowledgements

We thank the Brashares Lab for comments on data analysis that greatly improved the work as did early discussions about the project with Harshad Karandikar. Both M.W.S. and G.V. received funding from the Prince Albert II of Monaco Foundation.

## Author contributions

All authors conceived the original ideas for this paper. M.W.S. led the data analysis with L.G., S.M., G.Z., G.V., and K.B.; M.W.S. also led the writing with K.B., A.S., W.X., E.T., and C.A.

## Competing interests

The authors declare no competing interests
