## [Peer Review File · Nature Communications]

REVIEWER COMMENTS

Reviewer #1 (Remarks to the Author):

This manuscript investigates the extent and frequency human dimensions are considered in wildlife restoration and rewilding programs. The authors review 305 case studies from the ICUN's Global Reintroduction Perspective Series to identify the circumstances in which human dimensions are most frequently incorporated into wildlife restoration programs. Using a series of univariate logistic regression models the authors then examine how the incorporation of human dimensions varies between taxonomic groups, locations, and decision-makers. Of the case studies examined, only 42% included human dimensions as a program goal or indicator of success. The authors conclude that incorporating human dimensions into reintroduction plans can help to improve program success.

Whilst there are several strengths to the manuscript including the clarity of writing, there is room for considerable improvement. My major concern with the manuscript relates to the apparent mismatch between the aims reported in the abstract and the results reported in the paper. The authors claim to have "evaluated whether deliberate inclusion of human dimensions.... improves the effectiveness of wildlife restoration efforts" (page 2 lines 31 – 33). This is certainly an important question to ask, and the data set available to the authors is suitable for such an analysis. Unfortunately, however, the bulk of the manuscript instead focuses on less novel findings that better reflect a content analysis. I strongly recommend the authors either reframe the abstract and introduction to more accurately reflect the methods and results reported, or modify the methods, results, and discussion sections to focus on this question.

A secondary concern I have is in regard to the shallow and narrow treatment of relevant literature. I have provided detailed notes, recommendations and references below.

Page 2 lines 75- 76: Human dimensions should be better defined here and also expanded to include the economic factors linked to conservation (e.g. cost of conservation actions, opportunity costs associated with land-use decisions).

Page 3 lines 78 – 84. Systematic conservation planning tools have also been used to identify where and how restoration programs should take place (e.g. Smith et al 2008, Joseph et al 2009, Andre et al 2022). The systematic conservation planning literature speaks to the assertion made by the authors that human dimensions should be incorporated into conservation programs and as such, the authors

should familiarise themselves with this body of literature to better support the arguments made in the introduction and discussion.

Page 4 lines 103 – 111: This material is better suited to the discussion section. The results section should also include the evidence that “the inclusion of human dimensions increased the probability of a positive outcome for the restoration of wildlife”.

Page 4 lines 127 – 130: How does reporting bias affect the nature of case studies included in the series? Are there systems in place to facilitate reporting from smaller organisations that may not have the administrative capacity to file reports? I suspect these issues may influence the results and should be controlled for in the analysis if possible or at least addressed thoroughly in the discussion.

Page 5 lines 142 – 144: Were programs only led by a single stakeholder? If not, how were multi-stakeholder projects classified? Furthermore, aren't programs lead by local communities and private landholders inherently more likely to include human dimensions as they are conducting in-situ rather than ex-situ programs?

Page 7 lines 179 – 182: It appears to me that the 6 strategies you identified can be grouped into direct program activities (education, engagement, enforcement) and indirect or secondary activities (economic benefits, social tolerance, cultural benefits) where changes in the latter group e.g. social tolerance, may be the result of education or engagement. It would be interesting to see if there are different levels of success depending on which strategy or strategies were used.

Page 8 lines 191 – 194: Effective conservation efforts require threat abatement or mitigation and indeed the IUCN guidelines for reintroductions and translocations recommend threats be identified and removed before programs are initiated. If conflict has been identified as a primary threat to a species can it not be expected that addressing conflict would be essential to any wildlife restoration program? Do the case studies published in the perspectives series show a change in the number of conflict mitigation strategies included in programs since the publication of the reintroduction guidelines in 2013?

Page 8 lines 199 – 2014: These results speak to an interesting pattern; human dimensions were more frequently included when species were in close proximity to tangible direct threats compared to species facing less tangible and more indirect threats. This is something that should be discussed in greater depth.

Page 11 lines 222 – 223: Does the increased inclusion of human dimensions not also align with the introduction of best practices as per IUCN guidelines?

Page 11 lines 234 – 238: This result would be better-situated upfront e.g between lines 179 – 180.

Page 11 lines 238 – 241: Stakeholder categorisation needs to be more clearly defined in the methods section or the supplementary information. Were stakeholder categorisations based on project leads or project involvement?

Page 13 lines 262 – 264: This statement oversimplifies significant advancements in conservation research and practice as well as global policy shifts (Millennium Development Goals, Sustainable Development Goals, Convention on Biological Diversity just to name a few).

Page 13 lines 264 – 271: This result speaks of a failure to adequately address the complexity behind human-wildlife coexistence and as such, simply 'emphasising' human dimensions is unlikely to be a straightforward way to increase the success of wildlife restoration outcomes. Careful consideration of program locations, costs, stakeholder involvement, and stakeholder motivations is needed to achieve positive outcomes (see Guerrero et al 2017, Mason et al 2018).

Page 16 lines 329 - 335: Existing methods for addressing risks and threats should be cited here (e.g Carwardine et al 2019). There are also several frameworks for incorporating human dimensions into conservation activities (Schwartz et al 2017).

Page 12 – 17: Given the authors' experience and expertise with case studies from the reintroduction perspective series, this section seems to miss a valuable opportunity to provide concrete recommendations on how restoration program planning and program reporting can be improved. Relating the results to broader frameworks and policies could serve to improve practice and broaden the impact of the manuscript.

Figure 1: This is a nice figure but could be improved by including the % values in the graphic or figure caption.

Figure 2: This colour scheme doesn't appear to be colour blind friendly or suitable for greyscale. If the authors wish to use a diverging colour pallet, I would suggest broadening the range and including the % values at the top of each bar.

Figure 3: The font size of axis ticks and labels should be increased to improve readability.

Figure 4: This figure could be more impactful if it was incorporated into a more sophisticated figure panel incorporating the elements shown in figures 1 and 2. A map displaying the geographical distribution of reported case studies would complement the existing text.

Figure 5: Was the 'effectiveness' of these programs based on self-reported outcomes or based on statistical evidence?

I commend the authors' efforts in addressing this complex and important question. I hope these comments serve as constructive criticism to elevate the manuscript so that the findings are of increased relevance to academics, policymakers, and practitioners.

References:

André, L. V., Chinain, M., Gatti, C. M., Liao, V., Van Wynsberge, S., Tedesco, P., & Andréfouët, S. (2022). A systematic prioritization approach for identifying suitable pearl oyster restocking zones following a mass mortality event in Takarua Atoll, French Polynesia. *Marine Pollution Bulletin*, 176, 113472.

Carwardine, J., Martin, T. G., Firn, J., Reyes, R. P., Nicol, S., Reeson, A., ... & Chadès, I. (2019). Priority Threat Management for biodiversity conservation: A handbook. *Journal of Applied Ecology*, 56(2), 481-490.

Guerrero, A. M., Shoo, L., Iacona, G., Standish, R. J., Catterall, C. P., Rumpff, L., ... & Wilson, K. A. (2017). Using structured decision-making to set restoration objectives when multiple values and preferences exist. *Restoration Ecology*, 25(6), 858-865.

Hagger, V., Dwyer, J., & Wilson, K. (2017). What motivates ecological restoration?. *Restoration Ecology*, 25(5), 832-843.

Joseph, L. N., Maloney, R. F., & Possingham, H. P. (2009). Optimal allocation of resources among threatened species: a project prioritization protocol. *Conservation biology*, 23(2), 328-338.

Mason, T. H., Pollard, C. R., Chimalakonda, D., Guerrero, A. M., Kerr-Smith, C., Milheiras, S. A., ... & Bunnefeld, N. (2018). Wicked conflict: Using wicked problem thinking for holistic management of conservation conflict. *Conservation letters*, 11(6), e12460.

Schwartz, M. W., Cook, C. N., Pressey, R. L., Pullin, A. S., Runge, M. C., Salafsky, N., ... & Williamson, M. A. (2018). Decision support frameworks and tools for conservation. *Conservation Letters*, 11(2), e12385.

Smith, R. J., Easton, J., Nhancale, B. A., Armstrong, A. J., Culverwell, J., Dlamini, S. D., ... & Leader-Williams, N. (2008). Designing a transfrontier conservation landscape for the Maputaland centre of endemism using biodiversity, economic and threat data. *Biological Conservation*, 141(8), 2127-2138.

Reviewer #2 (Remarks to the Author):

Thanks for the opportunity to review the submitted manuscript, *Incorporating Human Dimensions Improves Wildlife Restoration Outcomes*. The submission appears to take a novel approach to understanding the impact of including “human dimensions” on the success of “wildlife restoration” as determined through an analysis of 305 case studies reported in the IUCN’s *Global Re-introduction Perspectives Series*. However, the authors’ current version of the manuscript could do more to fully deliver on all the promise(s) it may hold.

This paper appears to have 3 main assertions: 1) including human dimensions (HD) will lead to successful outcomes for reintroduced species, 2) that the inclusion of HD is predicated on 4 assertions, and 3) that HD inclusion will have increased over time. The first seems noteworthy, though could be developed further. The second seems tautological, as 3 of the 4 assertions (b-d of their prediction) are human dimensions (e.g., threats to wildlife by humans, human-wildlife conflict,

and local stakeholders playing a role) – thus, I’m unclear what benefits these findings actually provide or if this is a matter of semantics (i.e., perhaps we have different definitions of ‘human dimensions’ – a bit hard to tell given the authors don’t necessarily define ‘human dimensions’ at any point in the introduction). As such, some of the findings (e.g., Table S3) were intuitive because they were using human factors (e.g., transportation, energy production/mining, agriculture/aquaculture) as predictors of including human dimensions. And the third assertion of HD inclusion increasing over time doesn’t particularly add much to the paper – information has proliferated, and new approaches often do as well --- what does this actually tell us? Have other approaches (particularly those detracting from success being possible) been lost as inclusion of human dimensions increases? For now, the authors simply leave this assertion of HD increasing as a ‘good thing’ because they found through their analysis that HD leads to successful outcomes for reintroduced species (their first assumption), but it’s unclear from this analysis what kinds of human dimensions lead to success. Perhaps the authors could go further and consider what kinds of HD interventions specifically lead to success and for which species or geographic reach, or something pertinent to how conservation is conducted differently worldwide (as the literature is ripe with examples of how the success of conservation efforts varies globally).

Though the methods (i.e., analyzing case studies of the same reporting mechanism) are useful for sake of comparability, the writing (e.g., discussion) could benefit from greater contextualization of findings with regard to other conservation case studies that may have more nuance and insight into ‘success’. Additionally, a more thorough discussion of how the data were sourced could be informative (e.g., do only large-scale, well-funded conservation organizations submit such reports, and are they more likely to have capacity to include HD or be more likely to be successful? Or, more broadly, what attributes must a case study have to be included, and what biases may be associated with their inclusion?). This draft also framed ‘wildlife restoration’ more broadly than what the case studies entail (which is only re-introduction, right?), and does not examine the reporting institution’s indications of why the case study may have been successful or not, deeming those reports as likely to be biased.

More broadly, the authors introduced new results in the discussion section (which likely should be reorganized so results are all in results), and I believe the discussion could more clearly connect back to the literature helping to contextualize findings.

I am also providing in a separate file (.docx) additional detailed feedback / comments that arose as I read through the document, though – as with any review – I may have misunderstood parts of the paper or approach and welcome the opportunity to see a revised version that may help correct such misunderstandings. I hope these points are useful as you consider next steps, and best of luck with the process!

Reviewer #3 (Remarks to the Author):

Thanks for this interesting paper that tries to quantify the benefits of incorporating human dimensions in wildlife restoration and suggests case studies that did were on average ~10% more likely to have positive outcomes. This is a major topic in conservation and increasing calls for more local community involvement in wildlife restoration projects. Providing some hard evidence behind these calls that this is beneficial would be an important contribution.

I found the paper interesting, but have some major and minor concerns about the work which I think would greatly improve the paper.

Major concerns

1. Restoration is a big nebulous term and this research only focuses on translocations or re-introductions based on the methods summary – so I am concerned this title overreaching the scope of the actual research. I would suggest rewording the paper to ‘Incorporating Human Dimensions Improves Wildlife Translocation/Re-Introduction Outcomes’.
2. There is little background to how the IUCN Global Re-Introduction Perspectives Series is compiled? How are case studies picked? What are the biases in the series in terms of where the case studies come from? What does potential publication bias mean for your findings and conclusions?
3. In terms of bias, what was the breakdown of positive and negative outcomes from case studies? Was there a clear bias towards positive outcomes? This would help to quantify the publication bias in the data used.
4. Are there any other databases that could have been used to add to this to expand the analysis and dataset – for example, looking at full texts of studies for restoration interventions on Conservation Evidence? Systematic reviews or maps? Or maybe Panorama: <https://www.iucn.org/resources/conservation-tools/panorama>? The structure of the data may be different but I imagine a simple protocol could be easily applied to assessing these other data sources. At the moment the dataset is limited to one source and one type of literature (case studies).

5. In the data collection section, I'm rather concerned there is very limited information on how positive or negative outcomes were identified and whether this was done in a repeatable way?

a. How did the authors assign case studies to positive or negative outcomes - i.e., did they use a protocol? What about neutral outcomes where there were no changes? Was this put under negative? An ordinal analysis could be undertaken if there were positive, negative, and neutral outcomes.

b. What if case studies reported both positive and negative outcomes (particularly given mixed effects are often common)? Were these counted separately?

c. How many authors took part and did they assess inter-rater reliability? Without this information it's impossible to say how repeatable this research is and therefore casts doubt on the validity of the findings and conclusions. I think ways to test consistency in ratings include kappa tests or intraclass correlation tests and would say this needs to be done before this is considered for publication.

6. Why did you conduct separate logistic regression models? Would it not be more appropriate to include all variables within the same model to test the relative importance of each variable and control for multiple testing.

Minor concerns

7. I would encourage that you make the code and data freely available for this research via a citeable platform like Zenodo.

8. Fig.1: there seems unnecessary components here. There is no need for the key on the right to be as large as it is as this information is already clear in the figure. A simple box showing HD included versus no HD included with one colour and different shading would be fine, or this could even be described in the legend without a key.

9. Fig.3: please make it clear that there has been jitter applied as this is a binary variable.

10. Fig.4 seems unnecessary as this is simply a graph of two means and error which is clearly communicated in the text.

11. Discussion: The ~10% more likely to have positive outcomes message is clear, but it would be good to make sure the reader understands more clearly what these outcomes actually are in the start of the discussion? It is briefly mentioned in the methods but might not be found there. I felt I only found out the outcomes were related to the population of the species at the end of the discussion (L346).

12. The discussion generally feels quite long and could be shortened – this may free up space for more detail in the methods.

Reviewer 1: Comments to the author

- 1) This manuscript investigates the extent and frequency human dimensions are considered in wildlife restoration and rewilding programs. The authors review 305 case studies from the ICUN's Global Reintroduction Perspective Series to identify the circumstances in which human dimensions are most frequently incorporated into wildlife restoration programs. Using a series of univariate logistic regression models the authors then examine how the incorporation of human dimensions varies between taxonomic groups, locations, and decision-makers. Of the case studies examined, only 42% included human dimensions as a program goal or indicator of success. The authors conclude that incorporating human dimensions into reintroduction plans can help to improve program success.

We thank the reviewer for their time and their constructive comments. We believe the reviewers' comments and feedback have led to a much stronger manuscript.

- 2) Whilst there are several strengths to the manuscript including the clarity of writing, there is room for considerable improvement. My major concern with the manuscript relates to the apparent mismatch between the aims reported in the abstract and the results reported in the paper. The authors claim to have "evaluated whether deliberate inclusion of human dimensions.... improves the effectiveness of wildlife restoration efforts" (page 2 lines 31 – 33). This is certainly an important question to ask, and the data set available to the authors is suitable for such an analysis. Unfortunately, however, the bulk of the manuscript instead focuses on less novel findings that better reflect a content analysis. I strongly recommend the authors either reframe the abstract and introduction to more accurately reflect the methods and results reported, or modify the methods, results, and discussion sections to focus on this question.

We thank the reviewer for this important comment. We agree that the abstract and introduction needed to better reflect the totality of our analysis. We now highlight the other components of our analysis, which focus on the determinants for when and how objectives related to human dimensions are implemented, in the abstract and introduction. We've also emphasized the main finding in the results section on lines 238-241.

- 3) A secondary concern I have is in regard to the shallow and narrow treatment of relevant literature. I have provided detailed notes, recommendations and references below.

Thank you for your input and citations. We have added more detail in both the introduction (see response #9, lines 79-81, 90-95) and discussion sections (see responses #9, #21, #23, #33).

- 4) Page 2 lines 75- 76: Human dimensions should be better defined here and also expanded to include the economic factors linked to conservation (e.g. cost of conservation actions, opportunity costs associated with land-use decisions).

We've added 'economic' and 'psychological' to our definition of human dimensions on line 77. Our definition reflects the definition given in Riley and Sandström 2013. We've also included

examples of what human dimension activities may entail on lines 79-81, as well as in the abstract.

Riley, S. J. & Sandström, C. Human Dimensions Insights for Reintroductions. 55–78 (2013).

- 5) Page 3 lines 78 – 84. Systematic conservation planning tools have also been used to identify where and how restoration programs should take place (e.g. Smith et al 2008, Joseph et al 2009, Andre et al 2022). The systematic conservation planning literature speaks to the assertion made by the authors that human dimensions should be incorporated into conservation programs and as such, the authors should familiarise themselves with this body of literature to better support the arguments made in the introduction and discussion.

We've included literature on conservation planning in both the introduction on lines 90-92: "Many groups working to reintroduce wildlife now integrate social and ecological information into their conservation plans to better predict areas of wildlife tolerance, potential conflicts, and the distribution of benefits to local communities³¹⁻³⁴." and in the discussion on lines 351-354: "Future translocations should utilize conservation planning tools that integrate both ecological and socioecological variables which have been found to better predict the expansion of recolonizing wildlife populations, the occurrence of human-wildlife conflict, and effective release sites for reintroduced individuals^{31,32,55}." Citations added include André et al 2022, McCann et al. 2021, Smith et al. 2008, Ditmer et al. 2022.

- 6) Page 4 lines 103 – 111: This material is better suited to the discussion section. *Nature Communications* requires that "The final paragraph [of the introduction] must begin with a phrase like "In this work" or "Here, we show", and contain a brief summary of the major results and conclusions of the current work, written in the present tense."

- 7) The results section should also include the evidence that "the inclusion of human dimensions increased the probability of a positive outcome for the restoration of wildlife". We've included evidence that the inclusion of human dimension related objectives increased the probability of a positive outcome for the restoration of wildlife in the results section on lines 256-265: "Overall, translocation efforts that included human dimensions objectives were significantly more likely to have a positive outcome (0.94; 95% CI=0.88-0.97) than the translocation efforts that did not include human dimensions objectives (0.85, 95% CI=0.79-0.89; $p < 0.01$) ($p = 0.01$; Fig. 4)."

- 8) Page 4 lines 127 – 130: How does reporting bias affect the nature of case studies included in the series? Are there systems in place to facilitate reporting from smaller organisations that may not have the administrative capacity to file reports? I suspect these issues may influence the results and should be controlled for in the analysis if possible or at least addressed thoroughly in the discussion.

We thank the reviewer for pointing out potential reporting bias. The IUCN Conservation Translocation Specialist Group solicits submissions from many different groups; any project conducting a conservation translocation is eligible for inclusion. While we haven't systematically

reviewed the size of each group, there are a wide range of participants including large-scale conservation organizations like Conservation International and the San Diego Zoo, but also smaller community-run and species specific organizations like the Mabula Ground Hornbill Project or the Andean Cat Alliance. Still, even if there is a bias, we think our results are telling if major, well-resourced, and modern conservation organizations are failing to incorporate human dimensions into their objectives and/or failing to partner with local stakeholders. Regardless, for full transparency we address the potential reporting bias in lines 426-430 in the discussion.

9) Page 5 lines 142 – 144: Were programs only led by a single stakeholder? If not, how were multi-stakeholder projects classified?

Most programs were led by multiple stakeholders. In fact, only 20% of case studies were led by a single stakeholder. We have included clarification in the Methods section on lines 161-165: “The group(s) or stakeholder(s) for each translocation were identified from the author affiliations and Project Summary of each case study and were classified as government, academic, zoo, non-profit, local community, private landowner, and private company. Stakeholder classifications were based on project involvement; therefore, many case studies included multiple groups.” In addition, there is more detail in Table S2. We analyzed stakeholder involvement in two separate analyses. The first analysis was a univariate logistic regression looking at the impact of stakeholder group type on the inclusion of a human dimension objectives. Here, case studies with multiple stakeholders had multiple entries in the model. Therefore there were 305 case studies, but 741 separate entries in the model. However, for our second analysis we changed the variable to a binary inclusion or exclusion of a local stakeholder. For both analyses, we found similar results suggesting that when local community groups were involved in the translocation, the projects had a higher probability of including human dimension objectives.

10) Furthermore, aren't programs lead by local communities and private landholders inherently more likely to include human dimensions as they are conducting in-situ rather than ex-situ programs?

Only about 60% of case studies that included local community group leaders included objectives that were related to human dimensions. While some translocations include ex-situ activities (e.g. captive breeding), all translocations include an in-situ component (i.e. reintroduction or reinforcement of individuals).

11) Page 7 lines 179 – 182: It appears to me that the 6 strategies you identified can be grouped into direct program activities (education, engagement, enforcement) and indirect or secondary activities (economic benefits, social tolerance, cultural benefits) where changes in the latter group e.g. social tolerance, may be the result of education or engagement. It would be interesting to see if there are different levels of success depending on which strategy or strategies were used.

We agree that a deeper understanding of what types of human dimension objectives lead to better outcomes is important. We conducted a separate analysis comparing the direct and indirect strategies as suggested by the reviewer, but found no significant difference in outcome between translocations that included direct vs. indirect human dimension objectives (direct, n =

72; indirect, n = 18; both, n = 37; $p > 0.05$ in all cases). To keep the manuscript streamlined and concise, we did not include this additional analysis. That being said, we think further distentagling the types of human dimension activities and their impacts would make a great follow up study (which we now note on lines 367-369)

12) Page 8 lines 191 – 194: Effective conservation efforts require threat abatement or mitigation and indeed the IUCN guidelines for reintroductions and translocations recommend threats be identified and removed before programs are initiated. If conflict has been identified as a primary threat to a species can it not be expected that addressing conflict would be essential to any wildlife restoration program? Do the case studies published in the perspectives series show a change in the number of conflict mitigation strategies included in programs since the publication of the reintroduction guidelines in 2013?

We agree with this point and decided to conduct an additional analysis. We conducted univariate logistic regression with a binary response variable representing threat abatement (removed all threats noted in the case study, removed some threats noted in the case study, or didn't remove any threats noted in the case study), and a categorical variable indicating whether the project was before or after the publication of the 2013 guidelines. We found no significant effect (In all cases, $p > 0.05$). We decided to not include it in the manuscript.

13) Page 8 lines 199 – 2014: These results speak to an interesting pattern; human dimensions were more frequently included when species were in close proximity to tangible direct threats compared to species facing less tangible and more indirect threats. This is something that should be discussed in greater depth.

We agree that this was an interesting pattern, and we have included an additional analysis looking at the effects of direct and indirect IUCN threats on the inclusion of human dimension related objectives (lines 212-217 and 274-281). While the influence of threat type appeared important in the univariate analysis looking across all threat types, inclusion of a direct vs. indirect threat type covariate in the multivariate model revealed it to be of little importance relative to other covariates (Table S4).

14) Page 11 lines 222 – 223: Does the increased inclusion of human dimensions not also align with the introduction of best practices as per IUCN guidelines?

This was a great suggestion and we included a new analysis specifically looking at whether translocation efforts were more likely to include objectives related to human dimensions following the release of the reintroduction guidelines in 2013 (lines 221-225 and 308-310). We found no significant effect..

15) Page 11 lines 234 – 238: This result would be better-situated upfront e.g between lines 179 – 180.

We moved this result to the first paragraph of the results starting on line 233.

16) Page 11 lines 238 – 241: Stakeholder categorisation needs to be more clearly defined in the methods section or the supplementary information. Were stakeholder categorisations based on project leads or project involvement?

We added more details on stakeholder classification (see response #13).

17) Page 13 lines 262 – 264: This statement oversimplifies significant advancements in conservation research and practice as well as global policy shifts (Millennium Development Goals, Sustainable Development Goals, Convention on Biological Diversity just to name a few).

We have bolstered our commentary on this topic on lines 333-340: “Over the last few decades, there have been significant calls to better link conservation goals to sustainability goals, as well as to human values and wellbeing⁴⁸⁻⁵⁰. Additionally, major national and international conservation initiatives like the Convention on Biological Diversity and California’s 30x30 Executive Order aim to center human dimensions in their respective frameworks^{51,52}. Even the IUCN Guidelines for Reintroductions and Other Conservation Translocations has important recommendations for evaluating the social feasibility and conducting socioeconomic risk assessments of translocations⁵³.”

18) Page 13 lines 264 – 271: This result speaks of a failure to adequately address the complexity behind human-wildlife coexistence and as such, simply ‘emphasising’ human dimensions is unlikely to be a straightforward way to increase the success of wildlife restoration outcomes. Careful consideration of program locations, costs, stakeholder involvement, and stakeholder motivations is needed to achieve positive outcomes (see Guerrero et al 2017, Mason et al 2018).

We’ve deleted this sentence and largely changed this paragraph (lines 332-343).

19) Page 16 lines 329 - 335: Existing methods for addressing risks and threats should be cited here (e.g Carwardine et al 2019). There are also several frameworks for incorporating human dimensions into conservation activities (Schwartz et al 2017).

Thank you for drawing our attention to these papers. We have included them on lines 407 and 333-335..

20) Page 12 – 17: Given the authors’ experience and expertise with case studies from the reintroduction perspective series, this section seems to miss a valuable opportunity to provide concrete recommendations on how restoration program planning and program reporting can be improved. Relating the results to broader frameworks and policies could serve to improve practice and broaden the impact of the manuscript.

We agree that concrete recommendations are vital for improving the inclusion of human dimensions in conservation efforts. In the revised manuscript, we now explicitly note that the literature is full of frameworks and guidelines for incorporating human dimension activities into translocations (lines 441-443). Still, many projects are not including these in their planning and implementation, and we speculate the problem may be cost (financial and time) and its perceived benefit on lines 438-441: “Tools and practices to better understand the human dimensions of translocations like stakeholder engagement or participatory approaches can be

both costly and time consuming, yet our study underscores their importance. While the literature is ripe with frameworks and guidelines that emphasize the need to include human dimensions, too few projects adequately invest in the critical human dimensions needed for long-term success^{35,77}.” We hope that by illustrating the importance of including human dimension objectives, future translocation projects will be more inclined to incorporate existing guidelines and recommendations.

21) Figure 1: This is a nice figure but could be improved by including the % values in the graphic or figure caption.

We've included the % values in the figure caption.

22) Figure 2: This colour scheme doesn't appear to be colour blind friendly or suitable for greyscale. If the authors wish to use a diverging colour pallet, I would suggest broadening the range and including the % values at the top of each bar.

Thank you for this suggestion. We've changed the color to be both color blind friendly and suitable for gray scale. We have also included the % values in the caption.

23) Figure 3: The font size of axis ticks and labels should be increased to improve readability.

We increased the font size of the axis ticks and labels.

24) Figure 4: This figure could be more impactful if it was incorporated into a more sophisticated figure panel incorporating the elements shown in figures 1 and 2. A map displaying the geographical distribution of reported case studies would complement the existing text.

Given that all of the information in figure 4 is already reported in the text (mean and error) we have deleted the figure as per the suggestion of Reviewer 3

25) Figure 5: Was the 'effectiveness' of these programs based on self-reported outcomes or based on statistical evidence?

We've deleted the word 'effectiveness' since these are just examples cited from various case studies in the series.

26) I commend the authors' efforts in addressing this complex and important question. I hope these comments serve as constructive criticism to elevate the manuscript so that the findings are of increased relevance to academics, policymakers, and practitioners.

Thank you so much for all of your constructive comments. They were truly appreciated.

Reviewer 2: Comments to the author

27) Thanks for the opportunity to review the submitted manuscript, Incorporating Human Dimensions Improves Wildlife Restoration Outcomes. The submission appears to take a novel approach to understanding the impact of including “human dimensions” on the success of “wildlife restoration” as determined through an analysis of 305 case studies

reported in the IUCN's Global Re-introduction Perspectives Series. However, the authors' current version of the manuscript could do more to fully deliver on all the promise(s) it may hold.

We thank the reviewer for their constructive comments and feedback.

28) This paper appears to have 3 main assertions: 1) including human dimensions (HD) will lead to successful outcomes for reintroduced species, 2) that the inclusion of HD is predicated on 4 assertions, and 3) that HD inclusion will have increased over time. The first seems noteworthy, though could be developed further. The second seems tautological, as 3 of the 4 assertions (b-d of their prediction) are human dimensions (e.g., threats to wildlife by humans, human-wildlife conflict, and local stakeholders playing a role) – thus, I'm unclear what benefits these findings actually provide or if this is a matter of semantics (i.e., perhaps we have different definitions of 'human dimensions' – a bit hard to tell given the authors don't necessarily define 'human dimensions' at any point in the introduction). As such, some of the findings (e.g., Table S3) were intuitive because they were using human factors (e.g., transportation, energy production/mining, agriculture/aquaculture) as predictors of including human dimensions. And the third assertion of HD inclusion increasing over time doesn't particularly add much to the paper – information has proliferated, and new approaches often do as well --- what does this actually tell us? Have other approaches (particularly those detracting from success being possible) been lost as inclusion of human dimensions increases? For now, the authors simply leave this assertion of HD increasing as a 'good thing' because they found through their analysis that HD leads to successful outcomes for reintroduced species (their first assumption), but it's unclear from this analysis what kinds of human dimensions lead to success. Perhaps the authors could go further and consider what kinds of HD interventions specifically lead to success and for which species or geographic reach, or something pertinent to how conservation is conducted differently worldwide (as the literature is ripe with examples of how the success of conservation efforts varies globally).

Thank you for this broad-level feedback. It was very helpful for our revision and we believe our manuscript is much improved. We have included more detail on these comments below, however we'd like to address a few points.

We understand why the reviewer read the probability of the inclusion of human dimensions when local stakeholders were involved as tautological. We take that point, and have made an important distinction between the two in the revised manuscript. We define inclusion of stakeholders as projects in which local stakeholders were involved in project planning or implementation, whereas we define inclusion of human dimension as projects that explicitly incorporate human dimensions in the project planning phase as either a Goal or Success Indicator. Importantly, only 60% of projects that included local stakeholders actually included human dimension objectives in the planning phase. To make the distinction between incorporating human dimensions and including local stakeholders more explicit, we have reworded 'inclusion of human dimensions' throughout the manuscript to 'inclusion of human dimension objectives' (defined on lines 153-156: "We defined setting human dimension

objectives as a binary yes/no based on whether the project explicitly included either a Goal or Success Indicator that related to any aspect of cultural, political, economic, social, or psychological considerations⁴⁵.”).

We now define human dimensions more clearly in the introduction on lines 76-78 (“...human dimensions, or the social, political, psychological, economic, and cultural components of conservation...can be either foundational (providing information needed to understand the local context and stakeholders) or functional (being directly applied to management issues”). In addition, we have moved up Fig 5. (now Fig. 1) to give the reader a better understanding of the human dimension objectives presented in the case studies.

Per our temporal analysis, we have contextualized the importance of understanding whether objectives related to human dimensions have increased over time on lines 333-340: “Over the last few decades, there have been significant calls to better link conservation goals to sustainability goals, as well as to human values and wellbeing⁴⁸⁻⁵⁰. Additionally, major national and international conservation initiatives like the Convention on Biological Diversity and California’s 30x30 Executive Order aim to center human dimensions in their respective frameworks^{51,52}. Even the IUCN Guidelines for Reintroductions and Other Conservation Translocations has important recommendations for evaluating the social feasibility and conducting socioeconomic risk assessments of translocations⁵³. These advancements have all likely led to the observed increase in reported human dimension objectives. Still, even in the most recent 2021 IUCN report, only 50% of reported translocations set human dimension-related goals or indicators.”

We agree that future studies should work to disentangle the effectiveness of various human dimension strategies (lines 367-369). Given that only so many case studies included human dimension objectives, the relatively small sample size per strategy unfortunately precluded our ability to evaluate the relative effectiveness of different types of human dimensions. As more case studies are published by the IUCN, this will be a great question to come back to. That being said, per Reviewer 1s suggestion, we conducted a new post-hoc analysis evaluating the effects of direct vs. indirect human dimension objectives (see response #15). We found no significant results.

29) Though the methods (i.e., analyzing case studies of the same reporting mechanism) are useful for sake of comparability, the writing (e.g., discussion) could benefit from greater contextualization of findings with regard to other conservation case studies that may have more nuance and insight into ‘success’.

This is a great point and we have included findings from other translocations not included in the report on lines 348-351 and 363-365.

“In Ireland, inadequate consideration and consultation of a rural farming community prior to the reintroduction of the white-tailed eagle (*Haliaeetus albicilla*) resulted in widespread poisoning and high eagle mortality⁵⁴.”

“In Chile, wildlife tourism of an increasing puma (*Puma concolor*) population has led to a sharp decline in support for the lethal control of pumas, the primary cause of their decline and extirpation throughout the region⁵⁸.”

30) Additionally, a more thorough discussion of how the data were sourced could be informative (e.g., do only large-scale, well-funded conservation organizations submit such reports, and are they more likely to have capacity to include HD or be more likely to be successful? Or, more broadly, what attributes must a case study have to be included, and what biases may be associated with their inclusion?).

We thank the reviewer for pointing out potential reporting bias. The IUCN Conservation Translocation Specialist Group solicits submissions from many different groups; any project conducting a conservation translocation is eligible for inclusion. While we haven't systematically reviewed the size of each group, there are a wide range of participants including large-scale conservation organizations like Conservation International and the San Diego Zoo, but also smaller community-run and species specific organizations like the Mabula Ground Hornbill Project or the Andean Cat Alliance. Still, even if there is a bias, we think our results are telling if major, well-resourced, and modern conservation organizations are failing to incorporate human dimensions into their objectives and/or failing to partner with local stakeholders. Regardless, for full transparency we address the potential reporting bias in lines 426-430 in the discussion.

31) This draft also framed 'wildlife restoration' more broadly than what the case studies entail (which is only re-introduction, right?)

We agree that wildlife restoration is too broad of a term to use for the manuscript. We have replaced 'wildlife restoration' with 'wildlife translocation' throughout.

32) and does not examine the reporting institution's indications of why the case study may have been successful or not, deeming those reports as likely to be biased.

We argue that our classification of population-level success is more consistent than the authors due to differences in definition of success, length in project, and species translocation (e.g. generation times) on lines 178-193: “In general, the success of wildlife translocations can be measured in multiple ways including changes to the population, impacts to the ecosystem, and knowledge gained from the project. In the IUCN Global Re-introduction Perspectives Series, all authors rate the success of the project from 'Highly Successful' to 'Failure'. However, there may be inconsistency in how the authors define success. Therefore, we classified the outcome of the project as positive or negative based on the outcome to the wildlife population reported in the Project Summary and Reason(s) for Success sections of the reports. Case studies that we classified as having a positive outcome reported widespread survival, reproduction, or population growth, whereas case studies classified as a negative outcome reported either widespread mortality or population extinction. We used a binary positive or negative outcome instead of each individual outcome to reduce bias from the species in the case study (e.g., differences in generation times) or project (e.g., length of project) which could greatly impact differences in the reported outcome (e.g., survival vs. reproduction). Further, the binary outcome also increased the repeatability in our assessment of the project since the differences between

positive (widespread survival, reproduction, or population growth) and negative (widespread mortality and population extinction) is clear.”

33) More broadly, the authors introduced new results in the discussion section (which likely should be reorganized so results are all in results),

We have moved these results from the Discussion to the Results Section.

34) and I believe the discussion could more clearly connect back to the literature helping to contextualize findings.

We have further contextualized our findings with the literature in the discussion on lines 333-340, 348-351, 351-354, and 363-365 (see responses to reviewer 1’s comments to the same effect as well: Response #21, #23).

35) I am also providing in a separate file (.docx) additional detailed feedback / comments that arose as I read through the document, though – as with any review – I may have misunderstood parts of the paper or approach and welcome the opportunity to see a revised version that may help correct such misunderstandings. I hope these points are useful as you consider next steps, and best of luck with the process!

Thank you! These comments and edits were extremely helpful.

Reviewer 2 attached comments (transcribed from the attached Word document):

36) Line 31: This is true for both wildlife and humans, but assuming you mean human only here

Yes; clarified in the Abstract: “Successful translocation often hinges on coexistence between humans and wildlife, which can be fostered by education programs, economic incentives, conflict reduction assistance, and other means. Yet not all translocation efforts explicitly incorporate such human dimensions.”

37) Lines 32-33: Probably fine since just examples, but this is a pretty narrow view of HD and doesn’t fully capture the example you have included below (of “active inclusion of local stakeholders”)

We have changed the wording in the abstract (and throughout the manuscript) to differentiate between human dimension objectives and the inclusion of local stakeholders.

38) Line 35: All HD or just your definition of HD?

We have defined human dimension objectives on lines 153-156: “We defined setting human dimension objectives as a binary yes/no based on whether the project explicitly included either a Goal or Success Indicator that related to any aspect of cultural, political, economic, social, or psychological considerations⁴⁵.”

39) Lines 35-36: How is ‘positive outcomes’ being defined?

We have more explicitly defined positive outcomes in the Methods section on lines 178-193: “In general, the success of wildlife translocations can be measured in multiple ways including

changes to the population, impacts to the ecosystem, and knowledge gained from the project. In the IUCN Global Re-introduction Perspectives Series, all authors rate the success of the project from 'Highly Successful' to 'Failure'. However, there may be inconsistency in how the authors define success. Therefore, we classified the outcome of the project as positive or negative based on the outcome to the wildlife population reported in the Project Summary and Reason(s) for Success sections of the reports. Case studies that we classified as having a positive outcome reported widespread survival, reproduction, or population growth, whereas case studies classified as a negative outcome reported either widespread mortality or population extinction. We used a binary positive or negative outcome instead of each individual outcome to reduce bias from the species in the case study (e.g., differences in generation times) or project (e.g., length of project) which could greatly impact differences in the reported outcome (e.g., survival vs. reproduction). Further, the binary outcome also increased the repeatability in our assessment of the project since the differences between positive (widespread survival, reproduction, or population growth) and negative (widespread mortality and population extinction) is clear." We have also clarified positive vs. negative outcomes in the Abstract.

40) Line 36: "active inclusion" – how, in what ways? When?

We have further defined participation of local stakeholders as the group(s) or stakeholder(s) for each translocation were sourced from the author affiliations and Project Summary of each study and were classified as government, academic, zoo, non-profit, local community, private landowner, and private company (Lines 161-165).

41) Line 37: Could be useful to provide example here to help reader understand an example of what you mean by 'positive outcomes'

Added further explanation of our binary positive/negative in the Abstract and revisit the explanation in the results: "We found including human dimension objectives or involving local stakeholders improved wildlife population outcomes (i.e., increased survival, reproduction, or population growth)...."

42) Lines 57-58: Maybe expand what you mean by this? Are you including habitat restoration in support of existing populations or just things like captive breeding for purposes of reintroduction/supplantation?

We have reworded "wildlife restoration" to "wildlife translocation" throughout and have included a definition on lines 56-59: "Wildlife translocation (here defined as the intentional movement of organisms from one site to another for the benefit of conservation) serves as an increasingly important tool to combat widespread declines in global biodiversity²⁻⁵."

43) Lines 59-64: Would be great to include some examples beyond countries marked by colonization

Thank you for this important point. We have included an example of Arabian oryx on lines 60-61: "High-profile wildlife translocation success stories include the reintroduction of Arabian oryx (*Oryx leucoryx*) throughout the Arabian Peninsula and the peregrine falcon (*Falco peregrinus*) throughout the United States⁶⁻⁸."

44) Lines 65-66: As well as distrust between stakeholders, loss of resources, loss of jobs, etc.

Added more context around the importance of success on line 65: “Translocation programs require considerable time and resources, and their failure can lead to distrust between stakeholders, the loss of resources, and even the extinction or extirpation of entire populations or species.”

45) Line 68: Next paragraph has spaces between last word and references – check formatting throughout

Thank you for pointing this out.

46) Line 70-71: Could be useful to state why

Provided more context on why human dimensions in conservation are important on lines 83-85: “Incorporating human dimensions may ultimately prove as important to achieving conservation goals – if not more important - than biological or environmental factors because most threats to wildlife are directly attributed to humans²².”

47) Lines 71-73: This doesn't really seem like a range – consider adding more examples throughout, as your next sentences does a good job noting that HD supports multiple taxa, but doesn't help the reader to know what all HD actually is

We expanded this sentence to include multiple examples on lines 87-90: “Examples include providing resources to protect livestock from wildlife, education programs in local communities and schools, media campaigns to influence attitudes towards wildlife, economic benefits for landowners living with wildlife, and legal enforcement against illegal wildlife trade.”

48) Line 72: Seems vague – can you elaborate?

This sentence was largely changed. See comment #51.

49) Line 98: Depending on how you are defining HD, this might be tautological (e.g., inclusion of people leads to inclusion of factors associated with people).

Thank you for flagging this. We have made an important distinction between stated human dimension objectives and the participation of local stakeholders in the translocation (see response #32).

50) Line 110: Might consider not using this word, as not all HD leads to improved outcomes; thus the proliferation of HD could feasibly result in some of the same issues (e.g., failed restoration outcomes) as not including HD at all. In short, you want to stick with your hypothesis – that HD has increased over time – and the results you found raises important new research questions (e.g., what kinds of HD efforts lead to successful outcomes, since not all efforts do?)

We agree and have deleted the word ‘promisingly’.

51) Lines 125-126: Could be of interest to note what these examples convey

We have clarified what the examples are on lines 142: “Participants in the series are given a blank template and a few examples of case studies from previous volumes to draw upon.”

52) Line 148: Is this supposed to be “genes & diseases”, or “genes, diseases”?
Genes & diseases. Fixed.

53) Lines 154-155: A bit ironic given you (as authors) are perceiving and reporting success in your own view – would that possibly introduce bias (though at least it would be consistent)?

Good point. We have clarified how our binary positive/negative outcome reduces bias across species and length of project, but also reduces the inconsistency on lines 178-193: “In general, the success of wildlife translocations can be measured in multiple ways including changes to the population, impacts to the ecosystem, and knowledge gained from the project. In the IUCN Global Re-introduction Perspectives Series, all authors rate the success of the project from ‘Highly Successful’ to ‘Failure’. However, there may be inconsistency in how the authors define success. Therefore, we classified the outcome of the project as positive or negative based on the outcome to the wildlife population reported in the Project Summary and Reason(s) for Success sections of the reports. Case studies that we classified as having a positive outcome reported widespread survival, reproduction, or population growth, whereas case studies classified as a negative outcome reported either widespread mortality or population extinction. We used a binary positive or negative outcome instead of each individual outcome to reduce bias from the species in the case study (e.g., differences in generation times) or project (e.g., length of project) which could greatly impact differences in the reported outcome (e.g., survival vs. reproduction). Further, the binary outcome also increased the repeatability in our assessment of the project since the differences between positive (widespread survival, reproduction, or population growth) and negative (widespread mortality and population extinction) is clear.”

54) Line 156: How does length of project or differences in species relate to success? This doesn’t seem like enough information to help a general reader understand why your definition of success is ‘better’ or more ‘rigorous’ than the people most familiar with the project.

We have tried to clarify what we mean by length of project (see point #57). The evaluation of some translocations are ongoing, and for projects of shorter duration, the project leaders may have only observed survival thus far. However, that does not mean that reproduction or population growth will not occur in the future. Therefore, our binary outcome better controls for this bias across project length.

55) Line 177: Usually need to spell out # if first ‘word’ in a sentence
Fixed.

56) Lines 179-182: Seems like introduction might need a bit more explanation of these kinds of ‘human dimensions’ --- e.g., doing so might help allow for further explanation of things

like 'increasing social tolerance' (how is that different from the other categories you've listed here? Are some prior to, during, or a product of the process?)

We have moved up our categorizations of strategies (as well as the figure with examples) to the Methods section.

57) Lines 188-189: What about reptiles? Fig. 1 makes fish and reptiles look nearly the same. The p value for this comparison was 0.05 (Table S1) and the confidence intervals overlapped 0. Therefore, we chose not to include it as significant.

58) Lines 194-199: Again, as written, this seems tautological (e.g., involvement of impacted stakeholders *is* a 'human dimension', right?). Can you expand on why it's not?

As noted above (point #42), we now make a clearer distinction between the inclusion of objectives related to human dimensions and the participation of local community groups. Human dimension objectives are explicitly stated in the Goals or Success Indicator sections in the IUCN reports, whereas involvement of local stakeholders was identified from the author affiliations and Project Summary sections.

59) Lines 199-204: Unsurprising given that all of these things have a human element, whereas the 3 that had the lowest predicted probability are not typically seen as having a 'human dimension'.

We agree that this wasn't too surprising. However, we did find in a subsequent analysis that whether a threat was directly related to humans was not a strong predictor of including human dimension related objectives.

60) Lines 207-210: If allowed, perhaps include numbers in diagram? Hard to tell proportions or totals from this image, though otherwise looks nice

We included percentages in the figure.

61) Lines 212-217: Are these categories mutually exclusive, meaning a study could only include one of any of these? Assumingly no. So, how does the reader know what the proportions are of including each? To me at least, this reads as a little over 60% of restorations efforts that included local community also included some 'human dimension' and most, if not all, of those led to some positive outcome (also, can there be more than one positive outcome per project [and both positive and negative outcome in the same project] or is a 'positive outcome' defined as success?)

These categories are not mutually exclusive (see Response # 13) and your reading of the figure is correct. We hope that the clarification in the Methods section (lines 161-164) and the sample sizes in Table S2 clarify the proportions for the reader.

The explanation for our binary positive outcome variable can be found on lines 178-193: "The success of wildlife translocations can be measured in multiple ways including changes to the population, impacts to the ecosystem, and knowledge gained from the project. In the IUCN Global Re-introduction Perspectives Series, all authors rate the success of the project from 'Highly Successful' to 'Failure'. However, there may be inconsistency in how the authors define

success. Therefore, we classified the outcome of the project as positive or negative based on the outcome to the wildlife population reported in the Project Summary and Reason(s) for Success sections of the reports. Case studies that we classified as having a positive outcome reported widespread survival, reproduction, or population growth, whereas case studies classified as a negative outcome reported either widespread mortality or population extinction. We used a binary positive or negative outcome instead of each individual outcome to reduce bias from the species in the case study (e.g., differences in generation times) or project (e.g., length of project) which could greatly impact differences in the reported outcome (e.g., survival vs. reproduction). Further, the binary outcome also increased the repeatability in our assessment of the project since the differences between positive (widespread survival, reproduction, or population growth) and negative (widespread mortality and population extinction) is clear.”

62) Lines 244-247: Led or leads? Looks like these CI overlap, so effect size of differences might be small, right?

We have removed this figure based on comments from other reviewers.

63) Lines 273-277: These are all results – seems out of place in the discussion.

We have moved this section to the results starting on line 243.

64) Lines 277-279: This sentence does not match the next one – education and outreach is generally considered one-directional (even if in conversation), meaning the knowledge holder disseminates information to the ‘knowledge-deficient vessel’ (I’m exaggerating language here). Thus, assessment of attitudes is NOT education or outreach (it’s actually social science) – nor is ‘evaluation of its effectiveness (that’s also social science, or more specifically, program evaluation). Also, education and outreach can INFORM people of the goals of restoration or ways to support the effort (which might result in smooth implementation), but there are a lot of social science techniques (that are NOT education) that can be employed to help identify the goals of restoration as part of a process, and can assess where things may ‘break down’ in that process, but again – the examples provided here are not specifically education and outreach.

Thank you for pointing out this important distinction. We have fixed this sentence on lines 356-359: “Education and outreach was the most commonly reported human dimension strategy incorporated in translocation project planning. Importantly, education and outreach can help introduce people to the species and the goals of the project, as well as influence the behavior of the general public⁵⁶.”

65) Lines 280-282: This is a good example of education

Thank you.

66) Lines 282-284: Why how? If this is a discussion, seems useful to explain (even if briefly)

Added an example on lines 361-365: “Other popular human dimension objectives include increasing social tolerance and providing economic benefits to aid biodiversity conservation efforts^{16,57}. In Chile, wildlife tourism of an increasing puma (*Puma concolor*) population has led

to a sharp decline in support for the lethal control of pumas, the primary cause of their decline and extirpation throughout the region⁵⁸.”

67) Lines 284-285: Keep in mind that all of this depends on the reporting – so it doesn't mean that enforcement wasn't done or that it's not effective, simply means that whoever did the reporting didn't focus on that topic. Had there been a section of the report titled “Enforcement”, nearly all likely would have spoken to this topic even when not contributing to success (the degree to which enforcement was possible given various criteria)

Good point. We included the word ‘reported’ on line 366.

68) Line 293: Do you really know this? I don't recall any of your analyses being about this, only that inclusion of efforts to increase social tolerance can improve restoration outcomes as you've defined ‘success’

Deleted ‘increase social tolerance and’

69) Line 294-295: Why is this ‘promisingly’? as written, you're suggesting that HD is more often included after problems arise rather than as a way to prevent the occurrence of those problems or to preventatively reduce the severity of problems that might occur in the future. Thus, this seems problematic, not promising.

Deleted ‘promisingly’.

70) Lines 301-303: What does the literature say about these things? Otherwise, this sentence is a bit repetitive/doesn't necessarily add new context

We have deleted this sentence. Instead, we end with a call to tie translocation funding to human-wildlife conflict mitigation (lines 379-381): “Effectively resolving human-wildlife conflict has been paramount to the success of conservation, and funding for already costly translocation projects should be tied to addressing human-wildlife conflict in addition to species and habitat restoration⁶⁰.”

71) Lines 302-303: I assume the examples were about mitigation, not about paying people to live with wildlife – so edit accordingly if my changes don't capture the full range of examples

We have deleted this sentence and added a new ending to the paragraph (see comment #74).

72) Lines 311: Or even wildlife or ecosystems more broadly

We included your suggestion on line 387: “In some cases, top-down approaches in wildlife conservation have led to the displacement of local people and increased economic inequality, while providing little to no benefit for local people or even wildlife or ecosystems more broadly^{63,64}.”

73) Lines 315-319: Repetitive – also ideal not to (re)introduce results in the discussion, but to contextualize what you found in relation to other studies on these topics.

Deleted.

74) Lines 321-322: This is where numbers would help – you supply the count for # of studies included in the reporting system, and yes it appears that the system has captured more restoration studies related to birds and mammals. However, it's harder to assess the proportionality of those studies. Mammals definitely looks more than any other category, but is birds actually that different proportionally than the other categories (specifically fish and reptiles)? Probably yes, but the figure looks like fish, reptiles, and birds all could be 1/3 or so of studies included HD and 2/3 did not. Help the reader know for sure instead of making them guess

We have added percentages to the figure caption

75) Lines 324-327: So what does the literature say about why amphibians are ignored or why HD has not traditionally been included?

A review on human dimensions and amphibian conservation speculates that there is less attention to amphibians in this field due to the lower perceived value of the taxa. We include their review on lines 401-403: “Even so, few amphibian restoration efforts planned for human dimensions, perhaps due to a lower perceived value of this taxa to natural ecosystems and society⁷¹.”

76) Lines 333-335: Technically, you don't know this --- what works for some taxa may not work the same for all taxa. Plus, this is a bit vague, idealistic. Can you help the reader know more precisely why this would work and what the literature says?

We agree and have deleted that sentence.

Reviewer 3: Comments to the author

77) Thanks for this interesting paper that tries to quantify the benefits of incorporating human dimensions in wildlife restoration and suggests case studies that did were on average ~10% more likely to have positive outcomes. This is a major topic in conservation and increasing calls for more local community involvement in wildlife restoration projects. Providing some hard evidence behind these calls that this is beneficial would be an important contribution.

I found the paper interesting, but have some major and minor concerns about the work which I think would greatly improve the paper.

Thank you for your feedback. They were very helpful for the revision, and we think our manuscript is stronger for it.

Major concerns

78) 1. Restoration is a big nebulous term and this research only focuses on translocations or re-introductions based on the methods summary – so I am concerned this title overreaching the scope of the actual research. I would suggest rewording the paper to

‘Incorporating Human Dimensions Improves Wildlife Translocation/Re-Introduction Outcomes’.

We agree that wildlife restoration is too broad of a term to use for the manuscript. We have replaced ‘wildlife restoration’ with ‘wildlife translocation’ throughout.

79) 2. There is little background to how the IUCN Global Re-Introduction Perspectives Series is compiled? How are case studies picked? What are the biases in the series in terms of where the case studies come from? What does potential publication bias mean for your findings and conclusions?

We thank the reviewer for pointing out potential reporting bias. The IUCN Conservation Translocation Specialist Group solicits submissions from many different groups; any project conducting a conservation translocation is eligible for inclusion. While we haven’t systematically reviewed the size of each group, there are a wide range of participants including large-scale conservation organizations like Conservation International and the San Diego Zoo, but also smaller community-run and species specific organizations like the Mabula Ground Hornbill Project or the Andean Cat Alliance. Still, even if there is a bias, we think our results are telling if major, well-resourced, and modern conservation organizations are failing to incorporate human dimensions into their objectives and/or failing to partner with local stakeholders. Regardless, for full transparency we address the potential reporting bias in lines 426-430 in the discussion.

80) 3. In terms of bias, what was the breakdown of positive and negative outcomes from case studies? Was there a clear bias towards positive outcomes? This would help to quantify the publication bias in the data used.

We have moved this section up in the results to be featured more promptly, and we state the number of positive and negative outcomes on lines 237-238: “Most projects resulted in a positive outcome (n = 272); approximately 11% (n = 33) reported a negative outcome.”

81) 4. Are there any other databases that could have been used to add to this to expand the analysis and dataset – for example, looking at full texts of studies for restoration interventions on Conservation Evidence? Systematic reviews or maps? Or maybe Panorama: <https://www.iucn.org/resources/conservation-tools/panorama>? The structure of the data may be different but I imagine a simple protocol could be easily applied to assessing these other data sources. At the moment the dataset is limited to one source and one type of literature (case studies).

We thank the reviewers for suggesting additional data sources. While we agree that in theory more examples from the literature would lead to a more robust dataset, we believe that the standardization of each case study in the IUCN Global Re-Introduction Perspectives Series is one of the strengths of the analysis. In particular, most examples from academic journals lack information on the goals and success indicators of the wildlife translocation. We sampled the first 10 case studies from the Conservation Evidence page titled “Translocate to re-establish or boost populations in native range”, and found that the journal articles are very inconsistent in their reporting. In particular, they typically don’t cite specific goals or success indicators of the translocation other than the establishment of a population. Therefore, we believe any comparison between the case studies from the IUCN Global Re-Introduction Perspective Series

and journal articles would lead to misleading conclusions. Further, while Panorama is a great resource and includes a little more standardization comparable to the IUCN Global Re-Introduction Perspectives Series, all but one wildlife translocation within the database are already included in our dataset.

82) 5. In the data collection section, I'm rather concerned there is very limited information on how positive or negative outcomes were identified and whether this was done in a repeatable way?

We have expanded on how we classified positive and negative outcomes on lines 194-210: "In general, the success of wildlife translocations can be measured in multiple ways including changes to the population, impacts to the ecosystem, and knowledge gained from the project. In the IUCN Global Re-introduction Perspectives Series, all authors rate the success of the project from 'Highly Successful' to 'Failure'. However, there may be inconsistency in how the authors define success. Therefore, we classified the outcome of the project as positive or negative based on the outcome to the wildlife population reported in the Project Summary and Reason(s) for Success sections of the reports. Case studies that we classified as having a positive outcome reported widespread survival, reproduction, or population growth, whereas case studies classified as a negative outcome reported either widespread mortality or population extinction. We used a binary positive or negative outcome instead of each individual outcome to reduce bias from the species in the case study (e.g., differences in generation times) or project (e.g., length of project) which could greatly impact differences in the reported outcome (e.g., survival vs. reproduction). Further, the binary outcome also increased the repeatability in our assessment of the project since the differences between positive (widespread survival, reproduction, or population growth) and negative (widespread mortality and population extinction) is clear."

Further, we have discussed repeatability on lines 195-200: "We classified human dimension objectives and wildlife population outcomes through a collaborative calibration process. First, each coauthor independently evaluated thirty case studies to identify broad classifications of human dimension strategies. Next, we worked together to synthesize and refine classifications to comprehensively cover all human dimensions reported. We then reviewed all projects using the classification framework ensuring consistency by discussing all potentially ambiguous classifications with the entire group."

83) a. How did the authors assign case studies to positive or negative outcomes - i.e., did they use a protocol? What about neutral outcomes where there were no changes? Was this put under negative? An ordinal analysis could be undertaken if there were positive, negative, and neutral outcomes.

There were only positive and negative outcomes. We elaborate on how we classified each case study (see comment #86).

84) b. What if case studies reported both positive and negative outcomes (particularly given mixed effects are often common)? Were these counted separately?

Our definition of positive and negative outcomes was based solely on the outcome to the population (extinction, high mortality, high survival rates, reproduction, or population growth). Therefore, a case study could only be positive or negative. We recognize that this is a narrow definition of success (see comment #86), but we were interested in testing whether human dimensions impacted the outcomes for the translocated wildlife population.

85) c. How many authors took part and did they assess inter-rater reliability? Without this information it's impossible to say how repeatable this research is and therefore casts doubt on the validity of the findings and conclusions. I think ways to test consistency in ratings include kappa tests or intraclass correlation tests and would say this needs to be done before this is considered for publication.

All authors took part in the assessment of case studies. We elaborate on how we calibrated our responses (see response # 86 or lines 195-200). Further, we chose a binary outcome for the project to increase our repeatability in our assessment. The opportunity for error is small since the difference between positive (widespread survival, reproduction, or population growth) and negative (widespread mortality and population extinction) is straightforward.

86) 6. Why did you conduct separate logistic regression models? Would it not be more appropriate to include all variables within the same model to test the relative importance of each variable and control for multiple testing.

We have since added a model that included all variables to test their relative importance. See lines 212-217 and 274-281, as well as Table S4.

Methods:

“Next, we examined how the inclusion of human dimension objectives (as a binary response) varied among taxonomic groups, threats to the species, existence of a local history of conflict with the species, and the groups involved in the translocation. We evaluated differences for each variable using Tukey’s post-hoc pairwise comparisons. To compare the relative importance of each variable, we then conducted a multivariate logistic regression with the inclusion of a human dimension objectives as the binary response and the taxonomic group, existence of a local history of conflict with the species, whether the translocation involved local community groups, and whether one of the listed IUCN threats was a direct human threat as covariates.”

Results:

“After we identified taxonomic groups, groups involved in the translocation, IUCN threats, and a local history of conflict as significant predictors of the inclusion of human dimension related objectives, we evaluated the relative importance of each significant predictor in a global model. Like the univariate model results, translocations of mammals, local history of conflict, and whether the translocation involved local community groups were all significant predictors of including human dimension related objectives (Table S4). However, the translocation of fish taxa and the presence of a direct human threat were no longer significant relative when considered in conjunction with the other variables ($p > 0.05$ in both cases; Table S4).”

Minor concerns

87) 7. I would encourage that you make the code and data freely available for this research via a citeable platform like Zenodo.

Great point. We published our data Zenodo and provided “Source Data” as part of the review process with *Nature Communications*.

88) 8. Fig.1: there seems unnecessary components here. There is no need for the key on the right to be as large as it is as this information is already clear in the figure. A simple box showing HD included versus no HD included with one colour and different shading would be fine, or this could even be described in the legend without a key.

We have removed the key and included a description in the caption.

89) 9. Fig.3: please make it clear that there has been jitter applied as this is a binary variable.

Great point. We added this to the caption.

90) 10. Fig.4 seems unnecessary as this is simply a graph of two means and error which is clearly communicated in the text.

We have removed this figure.

91) 11. Discussion: The ~10% more likely to have positive outcomes message is clear, but it would be good to make sure the reader understands more clearly what these outcomes actually are in the start of the discussion? It is briefly mentioned in the methods but might not be found there. I felt I only found out the outcomes were related to the population of the species at the end of the discussion (L346).

Good point. We have added more context on lines 324-325.

92) 12. The discussion generally feels quite long and could be shortened – this may free up space for more detail in the methods.

We have rewritten large portions of the Discussion section and hope that the reviewer finds the content more streamlined and focused. We have also removed some information that was better suited for the Results section, as we added more detail to the methods to address the helpful feedback from you and the other reviewers.

REVIEWER COMMENTS

Reviewer #1 (Remarks to the Author):

Thank you for the opportunity to review the revised submission “Incorporating Human Dimensions Improves Wildlife Translocation Outcomes”.

Since initial submission the authors have reframed the introduction and improved the conceptual grounding of the paper through a more thorough treatment of relevant literature. The authors have also improved the manuscript by highlighting the gap between theoretical best practice and actions.

In addition to these changes, the authors have also attempted to clarify several key terms questioned by myself and other reviewers. For example a clearer definition of “Human Dimensions” is now provided on lines 79 – 81, 156 – 159, and there is an improved explanation for how project stakeholders were classified (161- 169).

Despite these commendable improvements, the manuscript requires revisions prior to publication.

Some statements regarding the scope and outcomes of the research are misleading. On lines 118 – 120 the authors state that this work is “the first to comprehensively evaluate how including human dimensions in the planning process impacts the effectiveness of wildlife translocations around the world”. Although the overall question asked by the authors is certainly suitable for such an analysis, the current manuscript does not include any form of impact evaluation.

Impact evaluations are a specific form of analysis that require confounding variables to be controlled for and counterfactual scenarios to be considered in order to determine a causal relationship between independent and dependent variables. This is a rapidly expanding area in conservation science and the Society of Conservation Biology is developing a rich resource base should the authors wish to adopt this methodological approach. Alternatively, the authors need to re-articulate the scope of the research.

My concern regarding reporting bias and how this was accommodated for in the analysis remains. I take the author’s point concerning reporting bias due to institutional capacity, but the fact that there is likely to be reporting bias due to positive and negative outcomes has not been adequately considered. There have been some interesting discussions on the effects of reporting bias on

conservation policy and practice that may assist in improving the analytical framework to address this (e.g. Wood 2020, Salafsky et al 2019, Spooner 2015).

In my opinion several aspects of the methodology including program classification and analysis (lines 178 – 193) still require further explanation. For example, if a project reported widespread survival but low population growth rates, was the program still classified as a ‘success’? Over what time frame was success measured and how was this standardised across taxa that presumably have varying intergenerational periods? Was there a higher probability of success in long term programs vs short term programs?

Without addressing these important factors (confounding variables, reporting bias, project time frames), it is very difficult to argue that there is a causal relationship between incorporating human dimensions into program plans, and program success. A lack of methodological detail also makes the approach hard to replicate or expand upon should additional data become available in the future.

Some more minor comments are also detailed below.

The predictors identified by the authors in line 109 – 114 are certainly interesting, but these predictions need to be underpinned by relevant literature and theory of change. A table or conceptual framework provided in the supplementary material would be useful to better support and justify these predictions, particularly for readers who are unfamiliar with translocations.

The structure of the discussion warrants further consideration. In the current format the topics jump around unnecessarily which detracts from the author’s key arguments.

For example, the paragraph discussing collaborative bottom-up approaches to conservation (lines 383 – 394) should be followed by the material concerning participatory approaches to including humans in wildlife conservation efforts (line 411 – 420).

This section would also be easier for readers to comprehend if it followed the same structure presented in the methods and results section (i.e. taxa, stakeholders, dimensions).

429: Do you mean Well-resourced rather than well-resource?

References:

<https://conbio.org/groups/working-groups/impact-evaluation/activities-and-resources/>

Salafsky, N., Boshoven, J., Burivalova, Z., Dubois, N. S., Gomez, A., Johnson, A., ... & Wordley, C. F. (2019). Defining and using evidence in conservation practice. *Conservation Science and Practice*, 1(5), e27.

Wood, K. A. (2020). Negative results provide valuable evidence for conservation. *Perspectives in Ecology and Conservation*, 18(4), 235-237.

Reviewer #3 (Remarks to the Author):

I thank the authors for revising their manuscript and the time taken to do so. I still have some concerns but I appreciate the authors responses and hope they improve the manuscript.

One concern I have is that the authors still do not seem to discuss the issues surrounding the publication bias in the literature they have studied. There is no discussion of how publication bias may have affected their results relating to the success of projects. There is clearly strong publication bias given that just 11% of case studies had a negative outcome - a reasonable guess would be that there is an equal chance of failure and success, perhaps even more of failure. Therefore, I think the limitations of literature studied need to be discussed more. For example, given we are missing a big proportion of the failures of translocation efforts, how robust are your findings that human dimensions increase conservation success? I am not disputing the result (it is unsurprising and intuitive), but I think there need to be more caveats around what you have found.

Also whilst I agree that human dimensions are very important and critical in most conservation projects, and I agree your findings show (L40-41) 'the absence of human dimension objectives in many translocations', I'm not sure your findings are (L40-41) 'underscoring their critical importance for conservation success.' A 10% difference in your binary measure of success is small but important, although I would not necessarily say it warranted the term 'critical'? The evidence presented does not support that strong claim. There are potentially other aspects of these case studies that could

have caused this difference that you have not controlled for - especially given the timeframe of the studies from 1960-2018. I would frame it more that in conservation we need to be looking for these marginal gains in success at every opportunity and this potentially provides one.

L323-325 - Again see my points above, but here you are implying causation when it would be more appropriate to say it was associated with 'an increased probability of ...'.

Figure 3 seems confusing in that the y axis mentions 'Restoration Efforts'. Shouldn't this be translocation or re-introduction?

Reviewer #1 (Remarks to the Author):

Thank you for the opportunity to review the revised submission “Incorporating Human Dimensions Improves Wildlife Translocation Outcomes”.

Since initial submission the authors have reframed the introduction and improved the conceptual grounding of the paper through a more thorough treatment of relevant literature. The authors have also improved the manuscript by highlighting the gap between theoretical best practice and actions.

In addition to these changes, the authors have also attempted to clarify several key terms questioned by myself and other reviewers. For example a clearer definition of “Human Dimensions” is now provided on lines 79 – 81, 156 – 159, and there is an improved explanation for how project stakeholders were classified (161- 169).

We thank the reviewer for their continued time and their constructive feedback. We believe the reviewers' comments and feedback have led to a much improved manuscript.

Despite these commendable improvements, the manuscript requires revisions prior to publication.

Some statements regarding the scope and outcomes of the research are misleading. On lines 118 – 120 the authors state that this work is “the first to comprehensively evaluate how including human dimensions in the planning process impacts the effectiveness of wildlife translocations around the world”. Although the overall question asked by the authors is certainly suitable for such an analysis, the current manuscript does not include any form of impact evaluation. Impact evaluations are a specific form of analysis that require confounding variables to be controlled for and counterfactual scenarios to be considered in order to determine a causal relationship between independent and dependent variables. This is a rapidly expanding area in conservation science and the Society of Conservation Biology is developing a rich resource base should the authors wish to adopt this methodological approach. Alternatively, the authors need to re-articulate the scope of the research.

We agree with the reviewers' sentiment about the scope of our research and acknowledge there may be other potential variables leading to the differences in project outcomes. In fact, we know that there are many ecological, environmental, and programmatic factors that can lead to successful translocations (see lines 72-74). We do not dispute the importance of those variables and many others. Rather, this study aims to analyze the relationship between the inclusion of human objective goals and the outcome of translocation efforts, since even minimal gains in success can lead to better conservation results. To address the concern, at the advice of both Reviewer 2 and 3, we have changed the language extensively by eliminating any reference to “impact” and removing language suggesting causality. We have changed the wording in the Abstract (lines 34-37, 39-41), Introduction (lines 102-106, 120-123, and 130-133), and Discussion (Lines 332-335, 339-343, 363-365, 459-460). We have pasted some key examples below.

Abstract (Lines 34-37)

“We found that fewer than half of all projects included human dimension objectives (42%), but that projects including human dimension objectives were associated with improved wildlife population outcomes (i.e., higher probability of survival, reproduction, or population growth).”

Introduction (Lines 102-106)

“To identify relationships between the inclusion of human dimension objectives in wildlife translocation efforts and program outcomes, we synthesized information from case studies reported in the International Union for Conservation of Nature (IUCN) Global Re-Introduction Perspective Series.”

Introduction (Lines 120-123)

“Our work is the first to correlate outcomes of wildlife translocations around the world with the inclusion of human dimensions in the project planning process. We found evidence that explicitly setting objectives related to human dimensions was associated with an increased probability of a positive outcome for the translocated wildlife population.”

Discussion (Lines 332-335)

“Human dimensions are increasingly thought to play a critical role in the success of conservation efforts, and our work supports this assertion by quantifying a strong relationship between the inclusion of human dimension objectives and the probability of success for wildlife translocation projects.”

Discussion (Lines 459-460)

“Therefore, analyses to understand even marginal gains in translocation success can be impactful for future conservation efforts.”

My concern regarding reporting bias and how this was accommodated for in the analysis remains. I take the author’s point concerning reporting bias due to institutional capacity, but the fact that there is likely to be reporting bias due to positive and negative outcomes has not been adequately considered. There have been some interesting discussions on the effects of reporting bias on conservation policy and practice that may assist in improving the analytical framework to address this (e.g. Wood 2020, Salafsky et al 2019, Spooner 2015).

We agree that our previous manuscript did not include sufficient detail on potential publication bias. We have addressed publication bias in the discussion on lines 441-451.

Discussion (Lines 441-451)

“Further, our results may be influenced by reporting bias against translocations conducted by smaller organizations as well as translocation failures. The publication rate for successful translocations is likely to be higher as many failed translocations are underreported which may partially account for the low failure rate (11%) in the IUCN report. Thus, our analysis is representative of the literature, but not all attempted translocations. Still, we’ve found that major, well-resourced conservation organizations and relatively overreported successful translocations

are failing to incorporate human dimensions into their efforts; this speaks particularly strongly to the overall lack of consideration for human dimensions if arguably the best-resourced and most successful translocations are foregoing important opportunities to improve conservation outcomes and local partnerships.”

In my opinion several aspects of the methodology including program classification and analysis (lines 178 – 193) still require further explanation. For example, if a project reported widespread survival but low population growth rates, was the program still classified as a ‘success’?

We have clarified the methodology on positive vs. negative outcome classification on lines 186-190. The success classification is based on a scale of widespread survival, reproduction, and population growth. Therefore, any evidence of widespread survival, at a minimum, would result in a successful classification. Widespread survival and low population growth rate would be reported as a success since both widespread survival and any evidence of population growth are markers for positive outcomes. We have also added a supplementary table (Supplementary Information 1; Table S2) with example excerpts from the case studies used to determine project outcome.

Methods (Lines 186-190)

“Case studies that we classified as having a positive outcome reported on a scale of widespread survival, reproduction, and/or population growth, whereas case studies classified as a negative outcome reported either widespread mortality or population extinction. Therefore, case studies only needed to report a minimum of widespread survival of the translocated individuals to be classified as a success.”

Over what time frame was success measured and how was this standardised across taxa that presumably have varying intergenerational periods? Was there a higher probability of success in long term programs vs short term programs?

There is a range in project lengths, as well as varying generation times depending on the species of interest. For this reason we chose to use a binary positive vs. negative outcome category. This way, projects could report a wide range of biological outcomes (widespread survival, reproduction, and/or population growth) and still be deemed successful. Thus, shorter term projects or translocations of species with slower generation times can still be considered successful if they report widespread survival. However, we have now revised our analysis to include both taxa and length of project as predictor variables in our analysis on how the inclusion of human dimension objectives impact the outcome (in the Methods on lines 211-215 and Results on lines 249-250). The inclusion of human dimension objectives is still significant, and neither taxa nor project length were significant ($p > 0.05$).

Without addressing these important factors (confounding variables, reporting bias, project time frames), it is very difficult to argue that there is a causal relationship between incorporating human dimensions into program plans, and program success. A lack of methodological detail also makes the approach hard to replicate or expand upon should additional data become available in the future.

As discussed above, we agree with this concern and have now changed the language to highlight a correlation rather than a casual relationship between human dimensions and outcomes in the Abstract (lines 34-37, 39-41), Introduction (lines 102-106, 120-123, and 130-133), and Discussion (Lines 332-335, 339-343, 363-365, 459-460). In addition, we included project time frames and taxa as predictor variables in our analysis, as well as a greater discussion on reporting bias in the Discussion. After accounting for these variables, our analysis still points to an important role of human dimensions in wildlife translocation success.

Some more minor comments are also detailed below.

The predictors identified by the authors in line 109 – 114 are certainly interesting, but these predictions need to be underpinned by relevant literature and theory of change. A table or conceptual framework provided in the supplementary material would be useful to better support and justify these predictions, particularly for readers who are unfamiliar with translocations. We agree and have added a table to the supplementary material (Supplementary Information 1; Table S1).

The structure of the discussion warrants further consideration. In the current format the topics jump around unnecessarily which detracts from the author's key arguments.

For example, the paragraph discussing collaborative bottom-up approaches to conservation (lines 383 – 394) should be followed by the material concerning participatory approaches to including humans in wildlife conservation efforts (line 411 – 420).

We thank the reviewer for this suggestion and have restructured the discussion to follow the suggested flow (taxa - stakeholders - dimensions).

This section would also be easier for readers to comprehend if it followed the same structure presented in the methods and results section (i.e. taxa, stakeholders, dimensions).

See comment above.

429: Do you mean Well-resourced rather than well-resource?

Yes, we changed “well-resource” to “well-resourced.”

Reviewer #3 (Remarks to the Author):

I thank the authors for revising their manuscript and the time taken to do so. I still have some concerns but I appreciate the authors responses and hope they improve the manuscript.

We thank the reviewer for their continued efforts and constructive comments. We believe our manuscript is much improved through these iterations.

One concern I have is that the authors still do not seem to discuss the issues surrounding the publication bias in the literature they have studied. There is no discussion of how publication bias may have affected their results relating to the success of projects. There is clearly strong publication bias given that just 11% of case studies had a negative outcome - a reasonable

guess would be that there is an equal chance of failure and success, perhaps even more of failure. Therefore, I think the limitations of literature studied need to be discussed more. For example, given we are missing a big proportion of the failures of translocation efforts, how robust are your findings that human dimensions increase conservation success? I am not disputing the result (it is unsurprising and intuitive), but I think there need to be more caveats around what you have found.

We agree that our previous manuscript did not include sufficient detail on potential publication bias. We have addressed publication bias in the discussion on lines 441-451.

Discussion (Lines 441-451)

“Further, our results may be influenced by reporting bias against translocations conducted by smaller organizations as well as translocation failures. The publication rate for successful translocations is likely to be higher as many failed translocations are underreported which may partially account for the low failure rate (11%) in the IUCN report. Thus, our analysis is representative of the literature, but not all attempted translocations. Still, we’ve found that major, well-resourced conservation organizations and relatively overreported successful translocations are failing to incorporate human dimensions into their efforts; this speaks particularly strongly to the overall lack of consideration for human dimensions if arguably the best-resourced and most successful translocations are foregoing important opportunities to improve conservation outcomes and local partnerships.”

Also whilst I agree that human dimensions are very important and critical in most conservation projects, and I agree your findings show (L40-41) 'the absence of human dimension objectives in many translocations', I'm not sure your findings are (L40-41) 'underscoring their critical importance for conservation success.' A 10% difference in your binary measure of success is small but important, although I would not necessarily say it warranted the term 'critical'? The evidence presented does not support that strong claim. There are potentially other aspects of these case studies that could have caused this difference that you have not controlled for - especially given the timeframe of the studies from 1960-2018. I would frame it more that in conservation we need to be looking for these marginal gains in success at every opportunity and this potentially provides one.

We agree that there are likely other factors impacting the success of wildlife translocations see lines 72-74). At the advice of both Reviewer 2 and 3, we have changed the language to better reflect the extent of our analysis in the Abstract (lines 34-37, 39-41), Introduction (lines 102-106, 120-123, and 130-133), and Discussion (Lines 332-335, 339-343, 363-365, 459-460). Additionally, we have reframed our findings as suggested by pointing to the importance of finding marginal gains in success for conservation (Lines 459-460).

Discussion (lines 459-460)

“Therefore, analyses to understand even marginal gains in translocation success can be impactful for future conservation efforts.”

L323-325 - Again see my points above, but here you are implying causation when it would be more appropriate to say it was associated with 'an increased probability of ...'.

Thank you for this comment. We have edited the text on line 337 accordingly.

Figure 3 seems confusing in that the y axis mentions 'Restoration Efforts'. Shouldn't this be translocation or re-introduction?

We changed the y-axis to say translocations.

REVIEWERS' COMMENTS

Reviewer #1 (Remarks to the Author):

Thank you to the authors for the time and effort revising the manuscript.

The revised submission addresses my previous concerns relating to reporting bias and causal assumptions through the inclusion of new analysis, substantially revised text, and the provision of supplementary tables. With these revisions, the authors clearly demonstrate the importance of including human dimensions in translocation program design and implementation.

I appreciate that the authors have re-structured the discussion section in this round of revisions, however given the strength of the abstract, introduction and methods sections, some further edits could be made to synthesise and strengthen the author's focal arguments. E.g. Line 433-436 "–Effectively resolving human-wildlife conflict has been paramount to the success of conservation. Translocation projects are costly and often risky, but explicitly addressing human-wildlife conflict could improve funding efficacy". This paragraph appears disjointed and incomplete with line 455 perhaps better suited for inclusion in the concluding paragraph.

Congratulations to the authors on this interesting and important study.

Reviewer #3 (Remarks to the Author):

Thanks for your revised manuscript and for taking into account all of my comments. I'm happy to recommend the publication of this manuscript subject to the satisfaction of other reviewers.

Reviewer #1 (Remarks to the Author):

Thank you to the authors for the time and effort revising the manuscript.

The revised submission addresses my previous concerns relating to reporting bias and causal assumptions through the inclusion of new analysis, substantially revised text, and the provision of supplementary tables. With these revisions, the authors clearly demonstrate the importance of including human dimensions in translocation program design and implementation.

Thank you so much for your continued feedback and comments. We are excited to publish this study.

I appreciate that the authors have re-structured the discussion section in this round of revisions, however given the strength of the abstract, introduction and methods sections, some further edits could be made to synthesise and strengthen the author's focal arguments. E.g. Line 433-436 "Effectively resolving human-wildlife conflict has been paramount to the success of conservation. Translocation projects are costly and often risky, but explicitly addressing human-wildlife conflict could improve funding efficacy". This paragraph appears disjointed and incomplete with line 455 perhaps better suited for inclusion in the concluding paragraph.

We agree. After re-reading the Discussion section we decided to delete this paragraph. The points made in the paragraph were all addressed elsewhere in the Discussion.

Congratulations to the authors on this interesting and important study.

Reviewer #3 (Remarks to the Author):

Thanks for your revised manuscript and for taking into account all of my comments. I'm happy to recommend the publication of this manuscript subject to the satisfaction of other reviewers.

Thank you once again for your constructive comments and feedback.